# Heat shock protein gp96 drives natural killer cell maturation and anti-tumor immunity by counteracting Trim28 to stabilize Eomes

Yuxiu Xu[1,5], Xin Li [1,5] ✉, Fang Cheng[1,2], Bao Zhao[3], Min Fang [1], Zihai Li [4] & Songdong Meng [1,2] ✉

The maturation process of natural killer (NK) cells, which is regulated by multiple transcription factors, determines their functionality, but few checkpoints specifically targeting this process have been thoroughly studied. Here we show that NK-specific deficiency of glucose-regulated protein 94 (gp96) leads to decreased maturation of NK cells in mice. These gp96-deficient NK cells exhibit undermined activation, cytotoxicity and IFN-γ production upon stimulation, as well as weakened responses to IL-15 for NK cell maturation, in vitro. In vivo, NK-specific gp96-deficient mice show increased tumor growth. Mechanistically, we identify Eomes as the downstream transcription factor, with gp96 binding to Trim28 to prevent Trim28-mediated ubiquitination and degradation of Eomes. Our study thus suggests the gp96-Trim28-Eomes axis to be an important regulator for NK cell maturation and cancer surveillance in mice.

Natural killer (NK) cells are innate lymphocytes that play a major role in immune surveillance[1–3]. They develop and start the process of maturation in the bone marrow and reach certain functional status when they migrate to the peripheral and differentiate into the mature status. The minimal cell population size and effector functions at the single cell level are required to launch an efficient anti-tumor NK cell response[4]. The expression of DX5 and Ly49 marks the maturation of NK cells. The NK cell maturation process includes three main discrete stages that can be discriminated by surface markers of CD11b and CD27. CD11b⁻CD27⁺ (referred to as CD27⁺ single positive, CD27 SP) are the less mature NK cells and give rise to CD11b⁺CD27⁺ (double positive, DP), which then differentiate into CD11b⁺CD27⁻ (CD11b⁺ single positive, CD11b SP) mature NK cells that also exist in the bone marrow[5]. In addition, KLRG1⁺ NK cells are at a highly mature stage of maturation[6]. Against tumor targets, mature NK cells (either DP or CD11b SP) are more cytotoxic than immature NK cells and express a distinct set of trafficking molecules that allow them to circulate in the blood[7]. Impaired NK maturation compromises tumor surveillance[8], while enhanced NK cell maturation increases host resistance to tumor growth[9,10].

NK cell development and maturation are orchestrated by a network of transcriptional factors (TFs)[11]. Among them, the transcription factor Eomesodermin (Eomes) is critical for NK cell maturation by supporting cell survival, resisting apoptosis, and promoting proliferation[12]. Eomes is characteristic of its highly homologous T-box DNA binding domains and a shared homology with T-bet, and large-scale chromatin accessibility analysis across immune subtypes predicted a major role of T-box TFs in the regulation of NK-cell specific enhancers[13]. Studies showed a strong correlation between Eomes and perforin expression in CD8⁺ T cells[14], suggesting the cytotoxic phenotype is driven by Eomes. Deleting Eomes in VAV1⁺ immune cells[15] in an inducible manner compromised NK cell development. Eomes depletion in NKp46⁺ cells showed that Eomes was required to preserve NK cell viability, especially at the CD27⁺CD11b⁺ stage, and that it was essential for cytotoxicity but not for IFN-γ secretion[13]. However, how Eomes itself is regulated remains unclear.

[1]CAS Key Laboratory of Pathogen Microbiology and Immunology, Institute of Microbiology, Chinese Academy of Sciences (CAS), Beijing, China. [2]University of Chinese Academy of Sciences, Beijing, P.R. China. [3]Department of Otolaryngology, The First Affiliated Hospital of Bengbu Medical College, Bengbu, Anhui 233004, China. [4]The Pelotonia Institute for Immuno-Oncology, The Ohio State University Comprehensive Cancer Center, Columbus, OH, USA. [5]These authors contributed equally: Yuxiu Xu, Xin Li. ✉e-mail: lix@im.ac.cn; mengsd@im.ac.cn

Glucose-regulated protein 94, also known as gp96 or Hsp90B1, is an endoplasmic reticulum (ER) resident member of the heat shock protein 90 (Hsp90) family involved in protein homeostasis[16]. Gp96 functions as a master ER chaperone to direct folding, assembly, maturation, or degradation of multiple cytoplasms and membrane proteins that are involved in regulating immune response and promoting cancer development, including EGFR (HER1)[17], HER2[18], Toll-like receptor[19], integrins[20], LRP6, GARP, and p53[21]. Depleting gp96 from specific cells reveals that gp96 plays an important role in the development and homeostasis of multiple immune cells, including regulatory T cells (Treg), B cells, CD4$^+$ T cells, and dendritic cells[20,22–24].

E3 ubiquitin ligase Trim28 is a member of the tripartite motif-containing proteins (Trim) family that functions as a corepressor of Kruppel-associated box zinc-finger factors[25]. Trim28 possesses a RING domain, two cysteine/histidine-rich motifs called B-box domains, and a coiled-coil domain. Trim28 can form MAGE-Trim28 E3 ubiquitin ligase complexes in cancer to target tumor-suppressor proteins such as 5′ adenosine monophosphate-activated protein kinase (AMPK) and p53 for ubiquitination and proteasome degradation[26,27].

In this study, we explore the effects of gp96 on NK cell development, maturation and function by using NK-specific gp96 knockout mice. Depletion of *gp96* results in impaired NK maturation and antitumor function. Our data further suggest that, under normal condition, the interaction between gp96 and Trim28 blocks the binding of Trim28 to Eomes and protects Eomes from ubiquitination and degradation for mediating downstream NK maturation. Our findings thus uncover a ubiquitin/lysosome machinery for Eomes degradation modulated by the gp96 regulatory network.

## Results

### NK-specific gp96 deficiency reduces NK cell maturation and function

To investigate the role of gp96 in regulating NK cell development and function, we generated *Ncr1*^Cre^*gp96*^fl/fl^ mice. These mice harbor a conditional deletion of *gp96*, specifically in NKp46$^+$ cells. Loss of gp96 protein in NK cells was verified by western blotting (Fig. 1A). The gp96-deficient mice contained significantly more NK cells in the spleen in the percentage of NK cells among total CD45$^+$ leukocytes, while NK cell percentage is significantly reduced in peripheral lymph nodes (pLN) and bone marrow (BM) (Fig. S1A). No such difference was observed in the liver. Other immune cells, including CD3$^-$CD19$^+$ B cells, CD4$^+$ T cells, CD8$^+$ T cells, CD11b$^+$Gr-1$^+$ myeloid, and CD11c$^+$ dendritic cells, display almost the same frequency in gp96 deficient mice as control WT mice (Fig. S1B).

We assessed NK cell maturation markers and found that compared to WT mice, a much higher proportion (-24.76 fold) of CD3$^-$CD122$^+$NK1.1$^+$DX5$^-$ immature NK cells (iNKs) and a significantly decreased proportion (-25.33 fold) of CD3$^-$CD122$^+$NK1.1$^+$DX5$^+$ mature NK cells were observed in the spleen of gp96-deficient mice (Fig. 1B). We observed that a subset of immature CD11b$^-$CD27$^-$ (double negative, DN) cells in DX5$^+$ NK cells apparently emerged in gp96-deficient mice. Compared to WT mice, gp96-deficient mice contained significantly more immature CD11b$^-$CD27$^-$ and less mature CD11b$^-$CD27$^+$ NK cells, and fewer CD27$^+$CD11b$^+$ and CD11b$^+$CD27$^-$ mature NK cells in spleen (Fig. 1B). Similar results were observed for NK cells in the bone marrow (Fig. 1C). Similar results were obtained between the analyses using CD3$^-$NK1.1$^+$ gate (Fig. S1C) and CD3$^-$NK1.1$^+$DX5$^+$ gate, so DX5$^+$ was used in an upstream gate. In addition, gp96-deficient NK cells expressed decreased maturation markers KLRG1, DX5 (Itga2), NKp46, and CD11b, both at the protein (Fig. 1D) and mRNA (Fig. 1E) levels.

To rule out any effects of the Ncr1-iCre allele on expressions of NK maturation markers, *Ncr1*^Cre^ mice were used to analyze the key readouts for flow cytometry. *Ncr1*^Cre^ and *gp96*^fl/fl^ mice showed similar expression levels of the marker molecules (Fig. S1D) and percentages of different maturation stages (Fig. S1E). Since NK and ILC1 share many common characteristics in terms of surface markers, we further analyzed ILC1 specific markers CD127 and CD49a expression in splenic DX5$^-$ NK cells examine if the defined iNKs contained ILC1s. As can be seen in Fig. S1F, G, a small fraction of cells expressed CD127 or CD49a, indicating that a minor amount of ILC1s were inadvertently classified as iNKs. However, there was no significant difference in the proportion of CD127$^+$ and CD49a$^+$ cells in CD3$^-$CD122$^+$NK1.1$^+$DX5$^-$ cells from *Ncr1*^Cre^, *gp96*^fl/fl^ and *Ncr1*^Cre^*gp96*^fl/fl^ mice, indicating that gp96 KO has no obvious effect on ILC1s.

Mixed bone marrow chimeras were constructed to evaluate the intrinsic effects of gp96 on NK maturation. Bone marrow cells isolated from CD45.1 WT and CD45.2 *gp96*$^{-/-}$ deficient mice were mixed at a 1:1 ratio and then adoptively transferred into sublethally irradiated recipient mice (Fig. 1F). The CD45.2$^+$ cells had higher immature and lower mature NK cell proportion than that from CD45.1$^+$ cells (Fig. 1G, H). These data collectively suggest that NK cell development in gp96 mice was blocked at the immature stage.

NK-specific gp96 knockout mice were more prone to tumor development in vivo than WT mice, as evidenced by significantly accelerated growth of murine LLC (Lewis lung carcinoma cell) (Fig. 2A) and MC38 colon carcinoma tumors (Fig. 2B), as well as increased endpoint MC38 tumor mass (Fig. 2C). These findings indicated that NK-specific gp96 deficiency promotes tumor growth in vivo. Moreover, compared to WT mice, reduced tumor-infiltrating NK cells were observed in MC38 tumors of gp96 KO mice (Fig. 2D–F), and these NK cells expressed lower degranulation marker CD107a (Fig. 2G), indicating a weakened cytotoxic function of tumor-infiltrating NK cells. Similar results were observed in the B16F10 melanoma model of tumor metastasis. Gp96 knockout mice exhibited more melanoma nodules (Fig. 2H, I) and lower frequency of IFN-γ$^+$ and CD107a$^+$ NK cells in the lung (Fig. 2G). We further tested the cytotoxic potential of gp96-deficient murine NK cells. Directly ex vivo, gp96-deficient NK cells had decreased cytotoxic activity against the prototypic YAC-1 target cells compared to WT NK cells (Fig. 2K). Additionally, in vivo killing experiments also demonstrated that the killing activity of NK cells from gp96 knockout mice was lower (Fig. 2L).

### Gp96 deficiency inhibits NK cell response to IL-15

To investigate the underlying mechanisms of gp96-mediated NK cell development and cytotoxicity, we set out to study whether gp96 regulates the response of NK cells to IL-15, which plays a critical role in NK cell maturation and effector functions[28]. Bone marrow cells of gp96-deficient mice were cultured in the presence of IL-15 with/without SCF. Unlike WT cells, *gp96*$^{-/-}$ bone-marrow cells failed to efficiently develop mature NK cells (Fig. 3A). Moreover, splenocytes from WT and gp96 KO mice were stimulated with different doses of IL-15 in vitro for 1 week. The frequency of the total NK cells among leukocytes was moderately higher in WT mice than that in gp96 KO mice (Fig. 3B), and notably, gp96 deficiency abrogated the increased effect of DX5$^+$ mature NK cells by IL-15 treatment as seen in WT NK cells (Fig. 3C). Low proportion of NK cells was detected from splenocytes even after 7 days of culture without IL-15 (Fig. 3C), which may be due to stimulation of trace amounts of autocrine IL-15 and other cytokines by splenocytes[29].

IL-15 promotes NK cell development by activating several downstream signaling pathways, including the JAK-STAT5 and PI3K-ATK-mTOR pathways[9,28]. Upon IL-15 stimulation, compared to WT cells, gp96-deficient splenic NK cells showed abruptly decreased phosphorylation of STAT5, as well as p-S6, a reliable marker of mTOR activation (Fig. 3D–F). Similarly, inhibition of gp96 with a specific inhibitor in NK92 cells also decreased the activation of STAT5 and mTOR (Fig. 3G, H).

### Gp96-deficient NK cells have specific and shared transcriptome signatures with Eomes-deficient cells

To further investigate the molecular characteristics of spleen NK cells from *Ncr1*^Cre^*gp96*^fl/fl^ mice, we performed 10 × genomics scRNA-seq. A

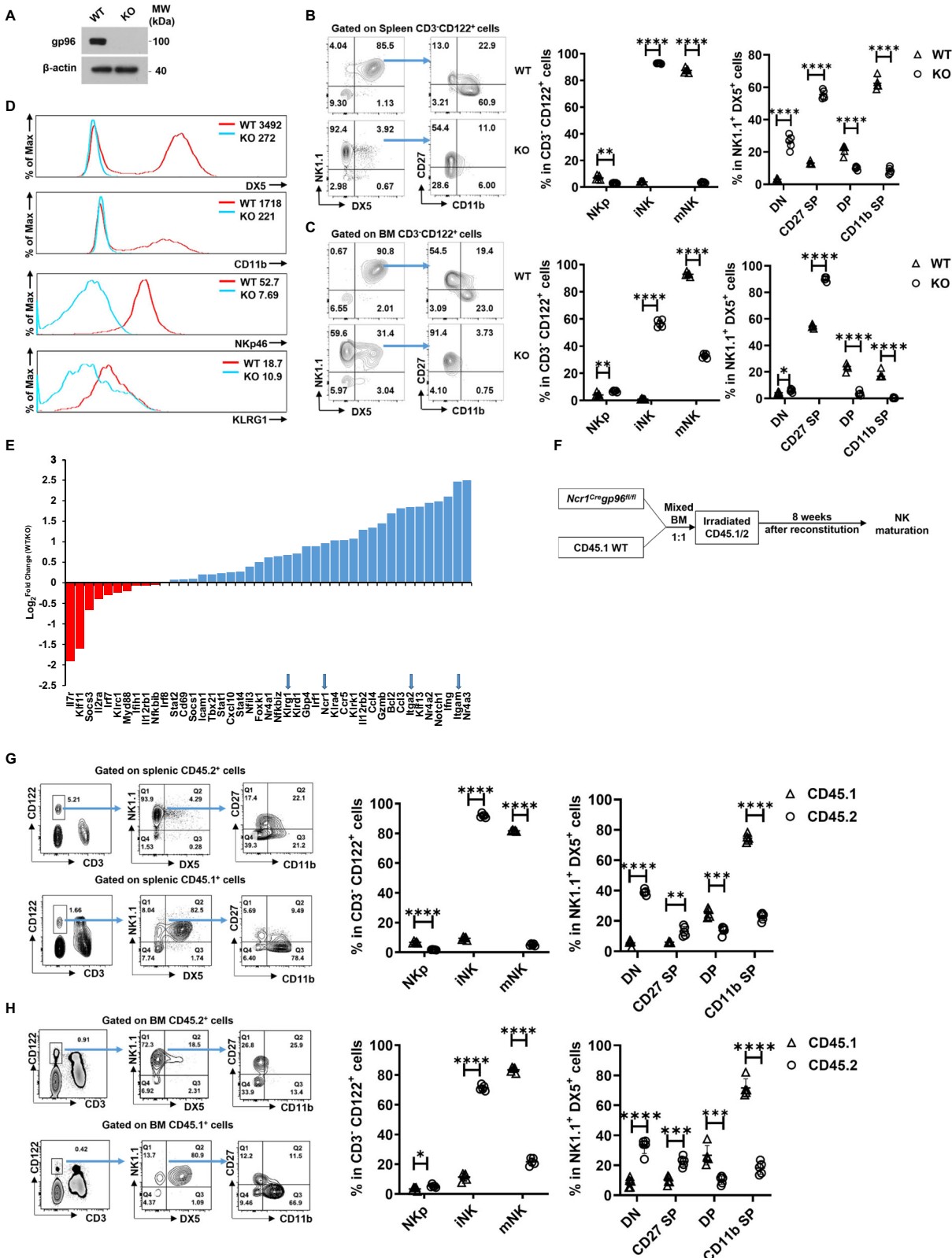

total of 15699 single NK cells (7349 WT and 8350 *Ncr1*^Cre*gp96*^fl/fl NK cells) were analyzed, and a mean of 1376 gene transcripts in WT and 1368 in *Ncr1*^Cre*gp96*^fl/fl NK cells was detected. Unsupervised clustering of all sequenced NK cells revealed six distinct clusters based on transcript signatures (Fig. 4A), with each cluster expressing unique gene markers (Fig. 4B). NK cells from *gp96*^−/− and WT mice exhibited distinct

cluster patterns (Fig. 4C and Fig. S2A). Of note, around 29% of the *gp96*^-/- NK cells were distributed in cluster 3, in comparison with only 3% of WT cells. In contrast, cluster 4 contained 0.3% of *gp96*^−/− and 18% WT NK cells. These clusters displayed key features in gene expression changes along NK cell maturation, including expressions of genes for effector molecules (Gzmb and Prf1) and genes for maturation markers

**Fig. 1 | NK-specific gp96 deficiency reduces NK cell maturation. A** Western blot of gp96 expression in splenic NK cells sorted from WT and KO mice. Representative flow cytometry plots showing the percentages of NKp (NK1.1⁻DX5⁻), iNK (NK1.1⁺DX5⁻), mNK (NK1.1⁺DX5⁺) cells gated on CD3⁻CD122⁺ splenocytes (**B**) and bone marrow (**C**) cells and representative flow cytometry plots showing the percentages of CD27⁻CD11b⁻ (DN), CD27⁺CD11b⁻ (CD27 SP), CD27⁺CD11b⁺ (DP), and CD27⁻CD11b⁺ (CD11b SP) cells on gated NK1.1⁺DX5⁺ splenocytes (**B**) and bone marrow (**C**) cells from WT and gp96-deficient mice. The numbers are percentages of the indicated quadrants among the gated cells. **D** Flow cytometry analysis of indicated marker levels. **E** Expression of indicated markers as determined by RNA-seq. The fold change indicates the difference in relative transcript expression between WT compared with *Ncr1*^Cre^*gp96*^fl/fl^ mice. **F–H** The chimeric mouse model was produced as in (**F**). Flow cytometry analysis of CD27 and CD11b on DX5⁺ NK cells from CD45.1⁺ cells and CD45.2⁺ cells in the spleen (**G**) and bone marrow (**H**). The data are representative of two independent experiments with similar results. Dots represent data from $n = 5$ mice/group. Mean ± SD is shown. Statistical significance was determined using two-tailed unpaired $t$ test. *$p < 0.05$, **$p < 0.01$, ***$p < 0.001$, ****$p < 0.0001$. $p$ values: (**B**) $p = 0.002$ (NKp), $p < 0.0001$ (iNK), $p < 0.0001$ (mNK), $p < 0.0001$ (DN), $p < 0.0001$ (CD27 SP), $p < 0.0001$ (DP), $p < 0.0001$ (CD11b SP), (**C**) $p = 0.009$ (NKp), $p < 0.0001$ (iNK), $p < 0.0001$ (mNK), $p = 0.0427$ (DN), $p < 0.0001$ (CD27 SP), $p < 0.0001$ (DP), $p < 0.0001$ (CD11b SP), (**G**) $p < 0.0001$ (NKp), $p < 0.0001$ (iNK), $p < 0.0001$ (mNK), $p < 0.0001$ (DN), $p = 0.0011$ (CD27 SP), $p = 0.0002$ (DP), $p < 0.0001$ (CD11b SP), (**H**) $p = 0.0281$ (NKp), $p < 0.0001$ (iNK), $p < 0.0001$ (mNK), $p < 0.0001$ (DN), $p = 0.0002$ (CD27 SP), $p = 0.0008$ (DP), $p < 0.0001$ (CD11b SP).

(CD11b/Itgam, Cma1 and Klrg1) and immature markers (Cd7, and Xcl1)[30] (Fig. 4D). Compared to cluster 3, cluster 4 exhibited higher expressions of Itgam, Klrg1, Cma1 and Cd27.

Next, we traced cell fate and reconstructed cell lineage direction using the RNA velocity approach. NK maturation in WT cells followed a single main branch (cluster 3/5 through 2 to cluster 0/1 and 4) without significant division. By contrast, velocity analysis revealed a different branch via cluster 3 through 4 to cluster 0/1 under gp96 knocking out condition (Fig. 4E), which is speculated that cluster 3 may be blocked in a relatively immature developmental stage. Since cluster 3 and cluster 4 are the different major clusters between WT and gp96 KO cells in both cell number and maturation process (Fig. 4C, E, Fig. S2D), we performed differential gene expression analysis among them. As seen in Fig. 4F, natural killer-mediated cytotoxicity was significantly enriched in cluster 4 ($P$.adj= $8.4 \times 10^{-9}$) but not significantly enriched in cluster 3 ($P$.adj = 0.076) by KEGG enrichment. Taken together, these results showed that NK-specific gp96 deficiency reduces its maturation and cytotoxic function.

To understand the molecular mechanisms whereby gp96 controls NK cell differentiation, we analyzed the expression of key transcriptional factors[11,31] in NK lineage development in gp96-deficient vs. WT NK cells. T-bet and Eomes play major roles in controlling NK cell maturation and function[11]. Using the mouse model expressing endogenously tagged Eomes[12], splenic NKs were isolated for ChIP analysis. WT and *Eomes*^-/-^ NK cells[12] were performed for RNA-seq to identify differentially expressed genes. Combination the ChIP-seq analysis with RNA-seq results was used to identify direct Eomes target genes. The overlap between Eomes-bound (ChIP-seq) and Eomes-regulated genes (RNA-seq) was identified as direct Eomes target genes. Analysis of the Eomes target genes revealed that the majority of upregulated transcripts in Eomes knockout NK cells were also upregulated in gp96 depleted NK cells, and most of the downregulated transcripts were downregulated simultaneously (Fig. 5A). By contrast, no such a trend was seen in T-bet knockout NK cells (Fig. S3A). In addition, genes differentially expressed between gp96 knockout and WT NK cells were correlated to those of *Eomes*^-/-^ and *Eomes*^+^ NK cells[32] (Fig. 5B). There was a significant decrease of Eomes levels in gp96-depleted NK cells compared to wild-type cells, as determined by flow cytometry (Fig. 5C and Fig. S3B) and western blot analysis (Fig. 5D). No obvious changes of Id2 and T-bet were observed under gp96 deficiency (Fig. S3B). Moreover, overexpression of Eomes largely rescued the effect of gp96 KO on DX5 expression in NK cells (Fig. 5E). These data indicate that gp96 may manipulate NK cell maturation mainly via Eomes.

#### Cellular gp96 binds to and protects Eomes from autophagy-lysosome-mediated degradation

Flow cytometry analyses show that expressions of gp96 and Eomes had a similar trend during NK maturation, which were high in CD11b⁺CD27⁺, and then decreased in CD27⁻ CD11b⁺ both in mouse spleen NK cells (Fig. 6A) and human PBMC NK cells (Fig. S4A). A confocal microscopy analysis further showed that cytoplasm gp96 levels were associated with NK maturation stages (Fig. 6B). The mRNA level of Eomes was not significantly reduced under gp96 depletion in sorted splenic NK cells (Fig. S4B, C), indicating that gp96 affects Eomes expression at the posttranscriptional level. Confocal microscopy results showed that gp96 colocalized with Eomes dominantly in the nucleus and minorly in cytosol but not in the ER of CD11b⁺CD27⁺ NK cells (Fig. 6C–E, and Supplementary Movie 1, 3).

Next, we explored the mechanism of gp96 KO-induced Eomes reduction. Depletion of *gp96* in splenic NK cells and HEK293 cells stably expressing Eomes caused a moderate decrease in Eomes stability (Fig. 6F, Fig. S4D), whereas ectopic expression of *gp96* caused an obvious increase in Eomes stability (Fig. 6G). Expression of Eomes was mostly increased in the presence of a lysosome inhibitor CQ, but not proteasome inhibitor MG132 in both primary NK and HEK293 cells (Fig. 6H), suggesting lysosome mediated degradation of Eomes. Moreover, the effect of gp96 on Eomes levels was abolished in the absence of *Atg5* (Fig. 6I, J), which plays a vital role in autophagosome formation by promoting the conversion of LC3-I to LC3-II[33]. *Gp96* KO facilitated Lys 63-linked, but not Lys 48-linked, polyubiquitin chains of Eomes protein (Fig. 6K, Fig. S4E) after autophagy inhibition with Baf A1. Since autophagy receptors play an essential role in selective autophagy as scaffolding proteins for recognizing ubiquitinated substrates, we determined whether the classical autophagy receptor SQSTM1 (P62) is involved in Eomes degradation. We found that the SQSTM1 interacted with Eomes, and the interaction was increased in the absence of *gp96*, and vice versa (Fig. 6L, Fig. S4F). Taken together, our results show that gp96 inhibits Eomes polyubiquitin and the subsequent autophagy-lysosome-mediated degradation.

#### Gp96 competes with E3 Ubiquitin Ligase Trim28 to interact with Eomes

Next, we screened for E3 ubiquitin ligase(s) targeting Eomes for lysosome degradation using immunoprecipitation and MS (Fig. S5A). Among the five candidate E3 ligases, silencing of Trim28 markedly and reproducibly led to upregulated protein expression of Eomes (Fig. 7A), and reduced Lys 63-linked, polyubiquitin chains of Eomes protein (Fig. 7B). Conversely, ectopic expression of Trim28 led to decreased Eomes protein stability in 293 cells (Fig. 7C). Moreover, overexpression of Trim28 in primary NK cells led to reduced Eomes levels (Fig. 7D) and increased Lys 63-linked-polyubiquitin (Fig. 7E). Interaction of Trim28 and Eomes was also confirmed by co-immunoprecipitation (Fig. 7F). We then dissected the potential Eomes-binding site (s) in Trim28 by employing ZDOCK 3.0.2 (https://zdock.umassmed.edu/).

There are several potential amino acid residues (Glu58, Leu62, His86, Ser87, and Val120) on the interface of the Trim28 Ring domain, which may be involved in interaction with Eomes through hydrogen bonds (Fig. 7G). We next mapped the Eomes-binding domain on Trim28 by constructing Trim deletion mutants (Fig. S5B). The Ring domain of Trim28 is essential for Eomes binding and E3 ubiquitin ligase activity, as demonstrated by co-immunoprecipitation (Fig. S5C).

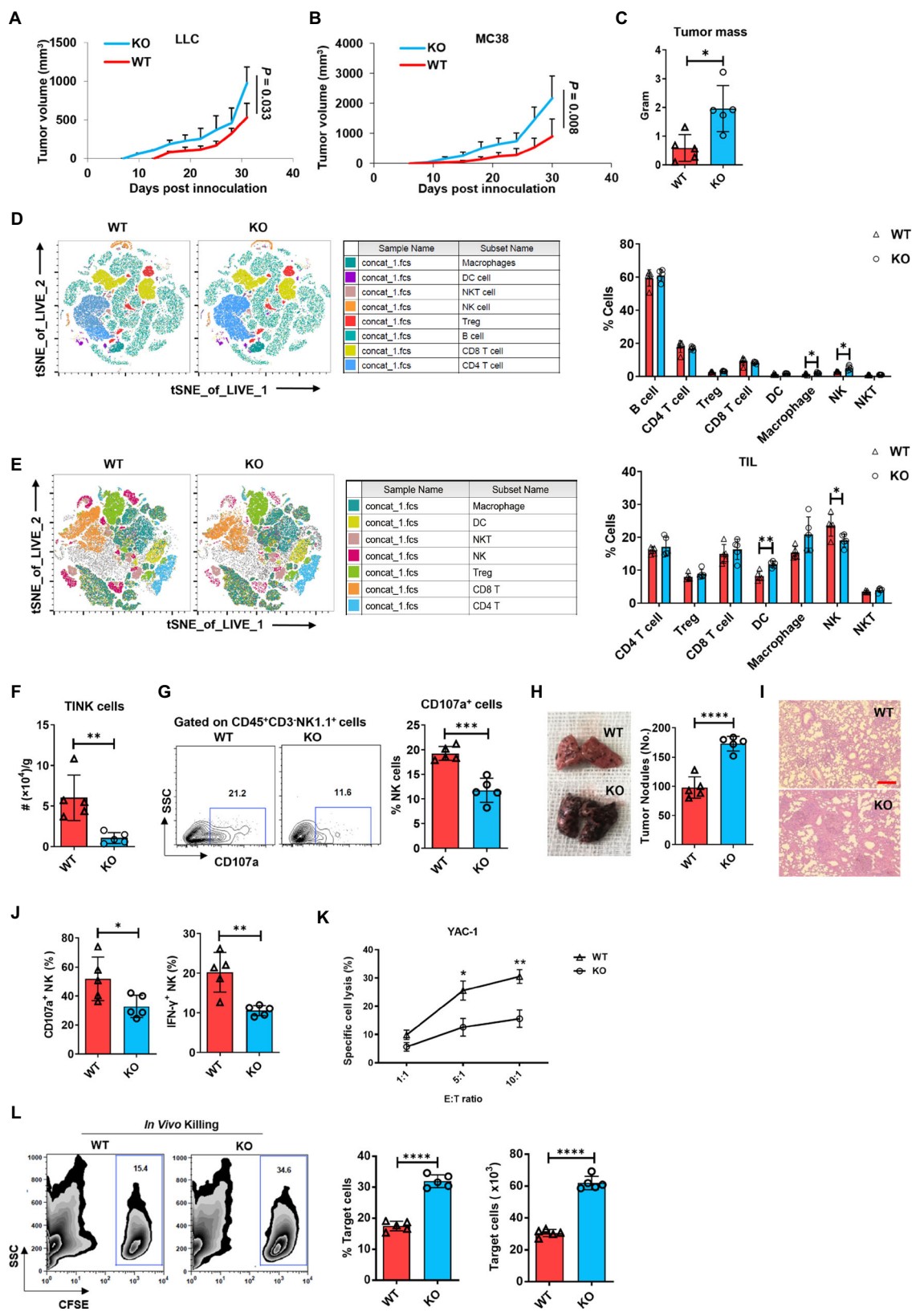

Expression of a Trim28 mutant with deletion of the defective Ring domain failed to downregulate Eomes expression (Fig. S5D). In keeping with this observation, Trim28, but not Trim28 truncate, significantly increased Lys 63-linked, polyubiquitin level of Eomes (Fig. S5E). These results indicate that Trim28 interacted and ubiquitinated Eomes protein by its Ring domain.

We then tested whether gp96 affected Trim28-mediated Eomes ubiquitination. Overexpression of gp96 in 293 cells led to decreased levels of Lys 63-linked-polyubiquitin of Eomes protein by Trim28, whereas depletion of gp96 increased these levels (Fig. 7H, I). Ectopic expression of gp96 hampered Trim28 binding to Eomes in HEK293 cells co-transfected with both Eomes and Trim28 (Fig. 7J). Increased

**Fig. 2 | NK-specific gp96 deficiency reduces NK cell anti-tumor functions.**
Female WT and KO mice were subcutaneously inoculated with LLC (**A**) or MC38 (**B**) tumor cells. Tumor size was measured every 2 days. **C** Endpoint tumor mass in mice 30d after MC38 cell inoculation. T-distributed stochastic neighbor embedding (tSNE) plots show the distribution of lymphocyte clusters in the spleen (**D**) and tumor tissue (**E**) as determined by flow cytometry analysis. **F** The absolute number of tumor-infiltrating NK cells from KO and WT mice as in (**E**). **G** Flow cytometry analysis of the frequency of cells expressing CD107a among tumor-infiltrating NK cells from MC38-bearing mice. **H** Representative lung nodules from WT and KO mice 2 weeks after intravenous (i.v.) injection of 0.25 × 10$^6$ B16F10 melanoma cells. **I** Hematoxylin and eosin (H&E) staining of lung sections from mice as in (**H**). Bar, 100 μm. **J** Frequency of cells expressing CD107a or IFN-γ among tumor-infiltrating

NK cells from mice as in (**H**), analyzed 14d after challenge. **K** Assessment of natural cytotoxicity of FACS-sorted splenic NK cells against YAC-1 target cells labeled with CFSE. **L** Frequency and the absolute number of remaining CFSE-labeled YAC-1 cells in the peritoneal cavities of KO and WT mice. CFSE-labeled YAC-1 cells were intraperitoneally injected and evaluated 24 h post-injection. The data are representative of two independent experiments with similar results. Dots represent data from $n = 5$ mice/group (**A–C, F–J, K**) and $n = 3$ mice/group (**D, E**). Mean ± SD is shown. Statistical significance was determined using two-tailed unpaired $t$ test. *$p < 0.05$, **$p < 0.01$, ***$p < 0.001$, ****$p < 0.0001$. $p$ values: (**C**) $p = 0.0106$, (**D**) $p = 0.0478$ (macrophage), $p = 0.0114$ (NK), (**E**) $p = 0.0016$ (DC), $p = 0.0275$ (NK), (**F**) $p = 0.005$, (**G**) $p = 0.0004$, (**H**) $p < 0.0001$, (**J**) $p = 0.0359$ (CD107a$^+$), $p = 0.0031$ (IFN-γ), (**K**) $p = 0.0236$ (5:1), $p = 0.0065$ (10:1), (**L**) $p < 0.0001$.

binding of gp96 to Trim28 led to decreased interaction of Trim28 with Eomes in a dose-dependent manner (Fig. S5F, G). In contrast, depletion of gp96 promoted Trim28 binding to Eomes both in primary mouse NK cells (Fig. 7K) and HEK293 cells transfected with Eomes and Trim28 (Fig. 7L). We further dissected the potential gp96-binding site(s) in Trim28. Interestingly, the Ring domain of Trim28 is also essential for gp96 binding, as evidenced by co-immunoprecipitation assay (Fig. 7M). There are two potential amino acid residues (Leu85, Cys121) involved in interaction with gp96 through hydrogen bonds on the interface of the Trim28 Ring domain, which is close in sequence to that of Eomes (Pro52, Leu82, and Gln123), as predicted by ZDOCK using Trim28 Ring domain (aa1-aa140) (Fig. 7N, O). These results suggest that gp96 and Eomes compete for Trim28 binding at the Ring domain, thus blocking Trim28-mediated ubiquitination and degradation of the Eomes protein.

Finally, we determined the interaction sites among gp96, Trim28 and Eomes in NK cells. As can be seen in Fig. 6E, Fig. S6A, B, both super-resolution immunofluorescence imaging and analysis of subcellular fractions showed that gp96 is mainly localized in the ER, and a small fraction of gp96 also exists in the cytosol and nucleus, which was consistent to previous studies[34,35]. While Eomes and Trim28 mainly exist within the nucleus, small amounts of them were found to be distributed in the cytosol. We therefore deduce that the interaction site among these 3 molecules may be the nucleus or the cytosol. The colocalization between Eomes and Trim28, or Trim28 and gp96 was confirmed in CD11b$^+$CD27$^+$ NK cells (Fig. S6C). Colocalization assays showed that an approximate 84 % of colocalization of gp96 and Trim28 was observed in cytosol as compared to 16 % in nucleus using the mask function of Imaris (Fig. S6D and Supplementary Movie 2 and 4). Z-stack images also show that only a small fraction of gp96 resides in cell nucleus relative to cytosol that includes cytoplasm as well as ER (Fig. S6E). Interaction between gp96, Eomes and Trim28 was further analyzed by immuno-precipitation from cytosol, nucleus and total membrane (including the ER and plasma membrane). As can be seen in Fig. S6F, interaction between gp96 and Trim28 was detected primarily in cell cytosol. The interaction between Eomes and gp96 (Fig. S6F) or Trim28 (Fig. S6G) was detected primarily in cell nucleus. ZDOCK analysis revealed that the binding site of gp96 in Eomes is at the interface (Arg296, Arg297, Arg284, Ser303, Ser370, Asn459, Tyr464) of the cleft formed by three helixes, and this partly overlaps with the binding site (Arg297, Gln438, Tyr464) of Trim28 (Fig. S6H). This indicates that interaction of gp96 with Eomes may sterically hinder the binding of Trim28 to Eomes.

In addition, gp96 KO had no obvious impact on Trim28 localization, but resulted in increased localization of Eomes in the cytosol (Fig. S7), which may be due to enhanced Eomes ubiquitination by gp96 KO and its subsequent translocation to the cytosol for autophagy-lysosome-mediated degradation.

## Discussion

NK cells are large granular lymphocytes that play an important role in innate immunity. In the current study, we utilized $Ncr1^{iCre}$ mice[36] to

generate NK cell conditional deletion of gp96. Using this cell-specific approach, we found that gp96 acts in multiple ways to regulate NK cell development and function. By comparing the percentages of NK (CD3$^-$NK1.1$^+$) in the spleen and bone marrow of gp96 WT and KO mice, our study shows that gp96 deficiency in NK cells affects its location in different organs. Compared to WT cells, gp96-deficient NK cells showed less maturation in the spleen and bone marrow (Fig. 1). In addition, gp96-deficient NK cells showed impaired killing function both in vitro and in vivo (Fig. 2). Significantly higher non-maturation markers (e.g., CD122, CD127) and lower maturation markers (e.g., DX5, NKp46) were observed in gp96 KO NK cells (Fig. 3). Also, the key transcriptional factor Eomes was significantly decreased in gp96-depleted NK cells. Finally, we showed that gp96 might promote Eomes expression by protecting them from ubiquitination and subsequent lysosome-mediated degradation. These results show that gp96 plays a critical role in NK cell development, maturation, and function via the regulation of Eomes (Fig. 7N).

The NK cell maturation process relies heavily on IL-15 signaling to support survival, stimulate proliferation, or antagonize apoptosis of NK cells[28]. This process depends on the presence of transcription factors T-bet, Eomes (Eomesodermin homolog), Nfil3, and Id2, among others, to direct the differentiation and expression of effector molecules[11]. Although the factors required for NK cell maturation have been extensively investigated, few regulators specifically targeting these transcriptional factors were reported. This study found that molecular chaperon gp96, a previously reported regulator of innate and adaptive immunity[19,20,22], correlated with the NK cell maturation process. We uncovered gp96 as a previously unidentified regulator in NK cell maturation in mice by modulating Eomes protein stability. Gp96 KO NK cells led to reduced STAT5 and S6 phosphorylation following ex vivo IL-15 stimulation (Fig. 3D–F). A significant reduction in the protein levels of Eomes was observed in NK cells from $Rptor$ cKO mice[37], which implies a possible regulation pathway among gp96 and Eomes via IL-15-pSTAT5 signaling that regulates the development of NK cells. In addition, we showed that gp96 expression also correlated with human NK cell maturation stages as analyzed by flow cytometry (Fig. S4A). These data suggest that human gp96 might have a similar role as mouse gp96 in promoting NK cell maturation. Cellular gp96 expression is controlled mainly by heat shock transcription factors (HSFs) and NF-κB, etc, and its differential expression is related to development, stress responses, inflammation, and cancer[38,39]. The exact regulatory pathway of gp96 expression and its impact on NK development and function both in physiological and pathological conditions deserves further investigation.

Eomes is critically important in a variety of different biological processes, disruption of which can lead to various human diseases[40]. It plays a vital role in the driving development of NK cells from CD27$^+$CD11b$^-$ stage to the CD27$^+$CD11b$^+$ stage[11,31]. In addition, Eomes is also required for induction of DX5 expression in NK cells[15]. We found that gp96 KO resulted in an obviously decreased DX5$^+$ mature NK cells by flow cytometry analysis (Fig. 1B, C), and overexpression of Eomes

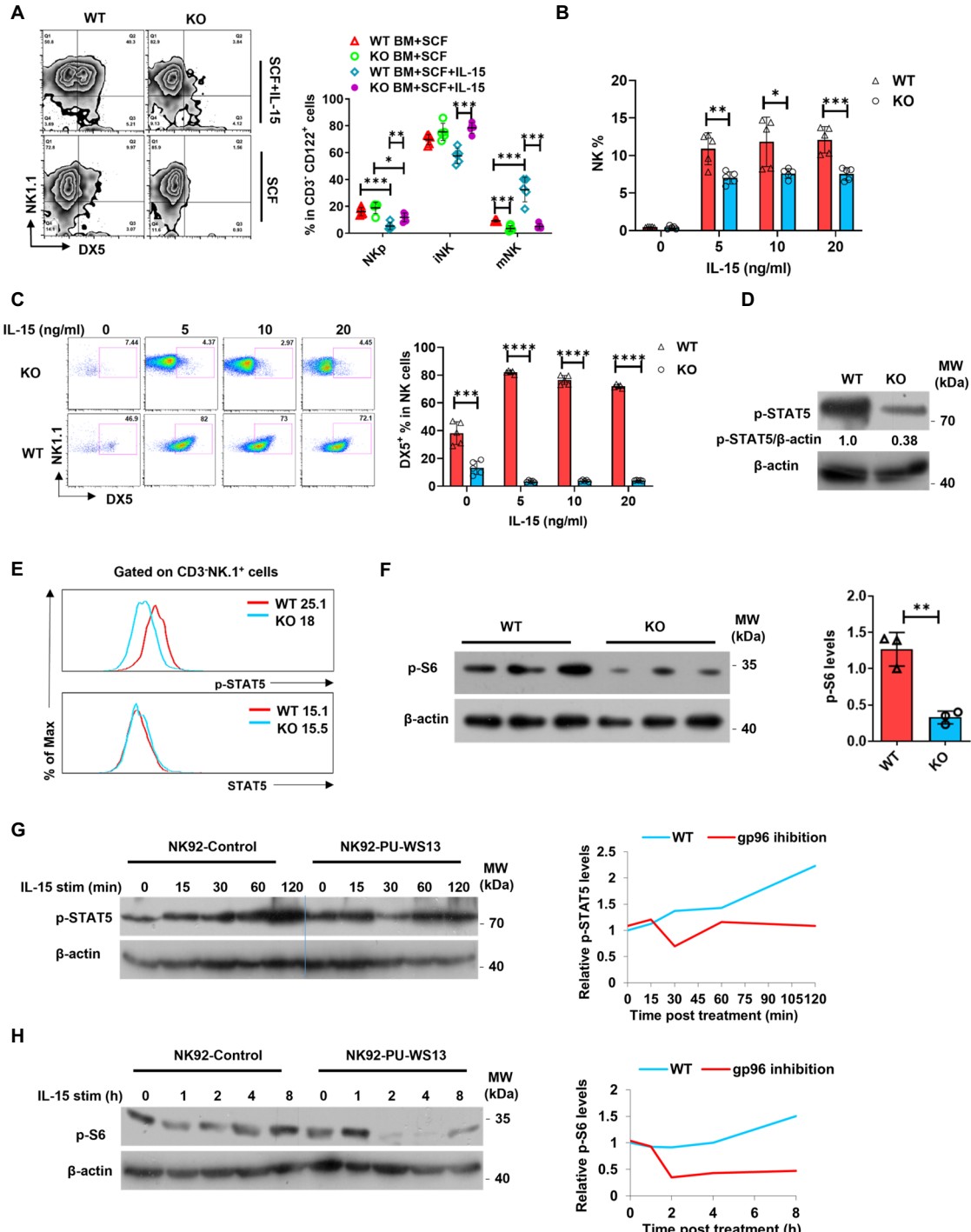

**Fig. 3 | NK-specific gp96 deficiency decreases NK cell response to IL-15. A** In vitro NK-cell differentiation from bone-marrow cells. Bone-marrow cells cultured with SCF in the presence (upper panel) or absence (lower panel) of 10 ng/ml IL-15 were subjected to the flow cytometric analysis. Frequencies of total NK cells (**B**) and DX5+ NK cells (**C**) among splenocytes treated with the indicated dose of IL-15 for 7 d. Primary NK cells were isolated and treated with 10 ng/ml IL-15 for detection of p-Stat5 (**D**) and p-S6 (**F**) levels. **E** Intracellular staining of p-Stat5 and Stat5 in splenic NK cells (gated on CD3⁻NK1.1⁺) after being stimulated with IL-15 (10 ng/ml). **G**, **H** NK92 cells were stimulated with IL-15 (10 ng/ml) after treatment with PU-WS13 or control. At indicated time points, cells were harvested and lysed, followed by

immunoblot using a phosphorylated (p)-Stat5, p-S6 antibody. The data are representative of two independent experiments with similar results. Dots represent data from $n = 5$ mice/group. Mean ± SD is shown. Statistical significance was determined using two-tailed unpaired $t$ test. \*$p < 0.05$, \*\*$p < 0.01$, \*\*\*$p < 0.001$, \*\*\*\*$p < 0.0001$. $p$ values: (**A**) $p = 0.0003$ (NKp, WT BM + SCF vs WT BM + SCF + IL-15), $p = 0.0168$ (NKp, KO BM + SCF vs KO BM + SCF + IL-15), $p = 0.0005$ (mNK, WT BM + SCF vs KO BM + SCF), $p = 0.0005$ (mNK, WT BM + SCF vs WT BM + SCF + IL-15), $p = 0.0002$ (mNK, WT BM + SCF + IL-15 vs KO BM + SCF + IL-15), (**B**) $p = 0.0052$ (5 ng/ml), $p = 0.0226$ (10 ng/ml), $p = 0.0008$ (20 ng/ml), (**C**) $p = 0.0004$ (0 ng/ml), $p < 0.0001$ (5 ng/ml), $p < 0.0001$ (10 ng/ml), $p < 0.0001$ (20 ng/ml), (**F**) $p = 0.0027$.

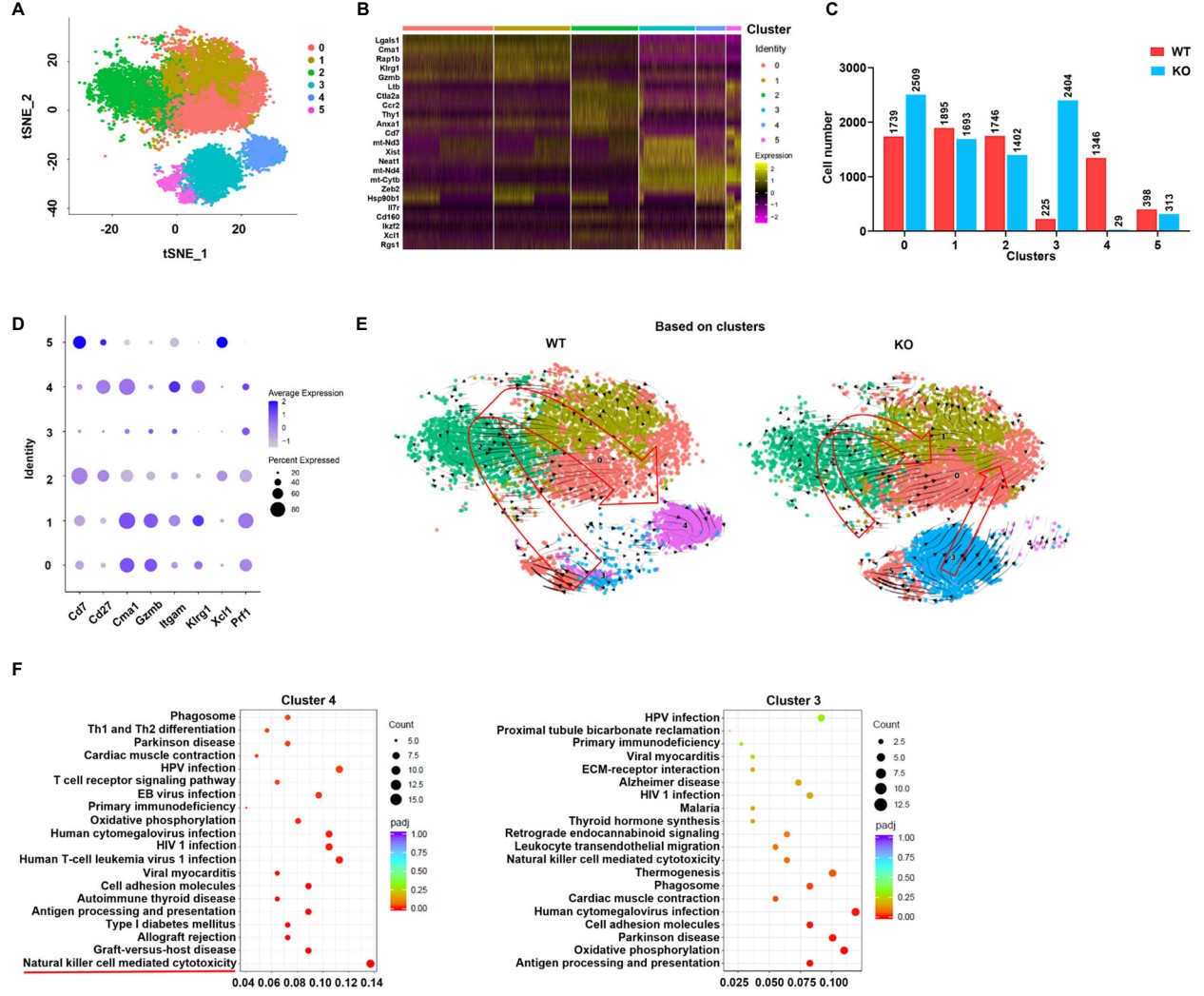

**Fig. 4 | High-throughput single-cell RNA-seq of NK cells reveals *gp96*^{−/−} clusters with distinct transcriptional signatures. A** A t-Distributed Stochastic Neighbor Embedding (tSNE) and graph visualization of the 15699 single NK cells defining 6 clusters. **B** Heatmap of marker genes in scRNA-seq clusters. Columns: single cells. Rows: cluster marker genes. Representative genes that are differentially expressed are on the left. **C** Cell number of *Ncr1*^Cre*gp96*^fl/fl and WT NK cells within each cluster. **D** Feature dot plot showing the relative expression levels of the indicated genes in

each cluster from (**A**). **E** Velocity analysis of the origin and direction of NK cell maturation. Velocity fields were projected onto the t-SNE plot. **F** KEGG analysis of DEGs for indicated clusters. Statistical significance was determined using one-sided Fisher's Exact Test with adjustments for multiple comparisons typically using methods like the Benjamini-Hochberg procedure. Selected KEGG terms with adjusted *P* values < 0.05 are shown.

could largely rescue DX5 expression (Fig. 5E). This indicates that gp96 KO-mediated Eomes degradation led to DX5 downregulation. Similarly, abruptly decreased NKp46 expression was observed in gp96 KO mice, which may be also due to Eomes reduction by gp96 KO as Eomes maintains the expression of NKp46 during NK cell development[15,41]. In addition, the significantly increased CD11b⁻CD27⁻ subset in gp96-deficient mice was the most immature NK cells, as this double-negative NK subset transit through less mature CD11b⁻CD27⁺ to CD11b⁺CD27⁺ and CD11b⁺CD27⁻ mature NK cells[42]. Meanwhile, knocking out gp96 in mice showed a similar NK cell development pattern to that of Eomes depletion, and specific and shared transcriptome signatures were observed between gp96-deficient and Eomes-deficient NK cells. In addition, analysis transcript signatures revealed that CD11b/Itgam and CD27 expressions in overall are relatively lower in cluster 3 and higher in cluster 4, in relative to other clusters (Fig. S2B). So there exits apparent overlap between cluster 3 and CD11b⁻CD27⁻ double-negative NK cells, both of which were increased by gp96 KO. So were for cluster 4 and CD11b⁺CD27⁺ DP NK cells, both of which were decreased under gp96 KO.

Therefore, we hypothesized that gp96, which acts as an important molecular chaperon, regulates NK maturation by influencing Eomes expression. Besides ER where gp96 majorly resides in, this chaperone may also be partly present in the cytoplasm and nucleus to interact with its client proteins[34,43,44]. Our data revealed that the E3 ubiquitin ligase Trim28 interacted with and promoted Eomes degradation via a selective autophagy-lysosome pathway. Moreover, in cell cytosol gp96 was found to primarily bind to Trim28 (Fig. S6D, F), and could compete for Eomes binding to Trim28, thereby protecting Eomes from Trim28-mediated degradation. As nucleus gp96 only constitutes a small fraction of the total cellular gp96 protein, and cytosolic gp96 includes cytoplasm and ER gp96, there are only subtle differences between total gp96 and cytosolic gp96 images, as indicated by arrows in Fig. S6D, E. While in cell nucleus, gp96 dominantly associates with Eomes (Fig. 6D, S6F), and may sterically block the interaction of Eomes and Trim28 (Fig. S6H). Indeed, increased cytosolic Eomes was observed in gp96 KO NK cells, which may be due to increased Eomes ubiquitination by Trim28 and subsequent translocation to the cytosol. Our results provide new insight into understanding the regulatory

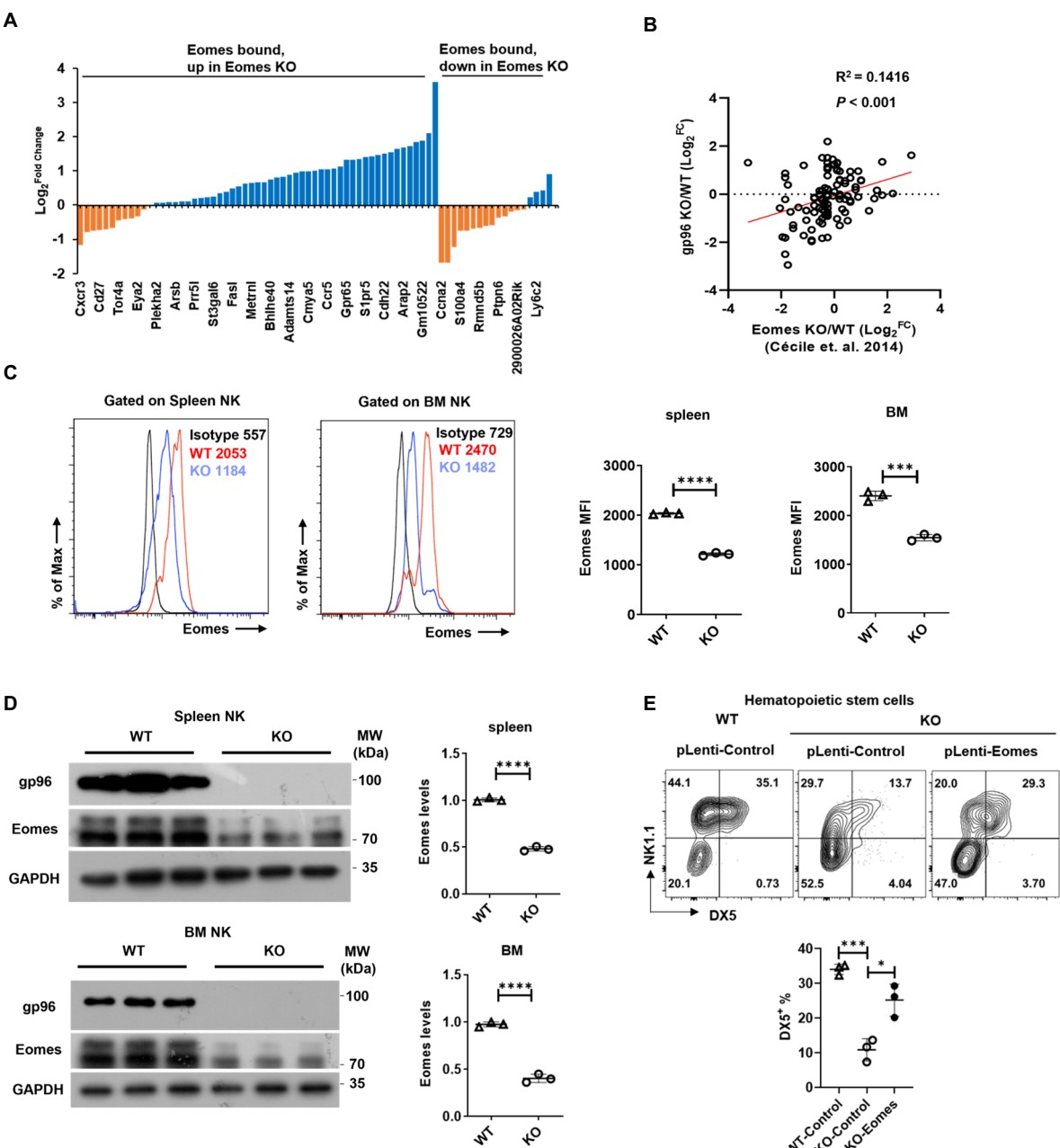

**Fig. 5 | Effect of gp96 on Eomes expression in NK cells. A** The fold changes indicated the difference in relative transcript expression of Eomes-bound genes in DX5⁺ splenic NK cells between WT compared with *Ncr1^Cre^gp96^fl/fl^* mice, as determined by RNA-seq. Y-axis showed Log2 fold changes of WT vs. KO mice. **B** Correlation analysis of Log$_2$ fold changes of Eomes-targeted genes between *gp96^−/−^* NK versus WT NK and *Eomes^−/−^* versus WT NK. **C** Flow cytometry analysis of levels of Eomes between *Ncr1^Cre^gp96^fl/fl^* and WT NK cells. **D** Western blot analysis of Eomes levels between *Ncr1^Cre^gp96^fl/fl^* and WT NK cells from spleen and bone marrow

(BM). **E** Flow cytometry analysis of DX5⁺ NK cell percentage in WT or gp96 KO BM hematopoietic stem cells infected with Eomes or control Lentiviral vector. Mean ± SD of three replicates is shown. Statistical significance was determined using two-tailed unpaired *t* test. *$p < 0.05$, ***$p < 0.001$, ****$p < 0.0001$. *p* values: (**B**) $p = 0.0004$, (**C**) $p < 0.0001$ (spleen), $p = 0.0002$ (BM), (**D**) $p < 0.0001$ (spleen), $p < 0.0001$ (BM), (**E**) $p = 0.0003$ (WT-control vs KO-control), $p = 0.0117$ (KO-control vs KO-Eomes).

mechanism of Trim28-mediated Eomes ubiquitination that was simultaneously affected by gp96 (Fig. 7P).

As gp96 may regulate DX5 expression via Eomes, in current study DX5⁺ was used as an upstream gate flow cytometry. However, much less immature CD11b⁻CD27⁻ NK cells and more DP/CD11b⁺ SP mature NK cells in gp96 KO mice were observed by using CD3⁻NK1.1⁺DX5⁺ gate than using CD3⁻NK1.1⁺ gate, as seen in Fig. 1B, S1C. This phenomenon was not seen in WT mice. Thus, using DX5⁺ as an upstream gate may miss certain types of NK cells that lose DX5 expression.

In summary, our findings demonstrate that gp96 as a chaperon is involved in regulating NK maturation and cytotoxic function. We further uncovered the mechanism of gp96-mediated increase of Eomes protein expression in NK cells, in which cellular gp96 bound to the E3 ubiquitin ligase Trim28 and sterically blocked the interaction between Trim28 and Eomes, thereby inhibiting Eomes ubiquitination and subsequent autophagic degradation. Our study, therefore, provides an avenue to target NK cells for viral clearance and tumor surveillance based on gp96 expression and activity.

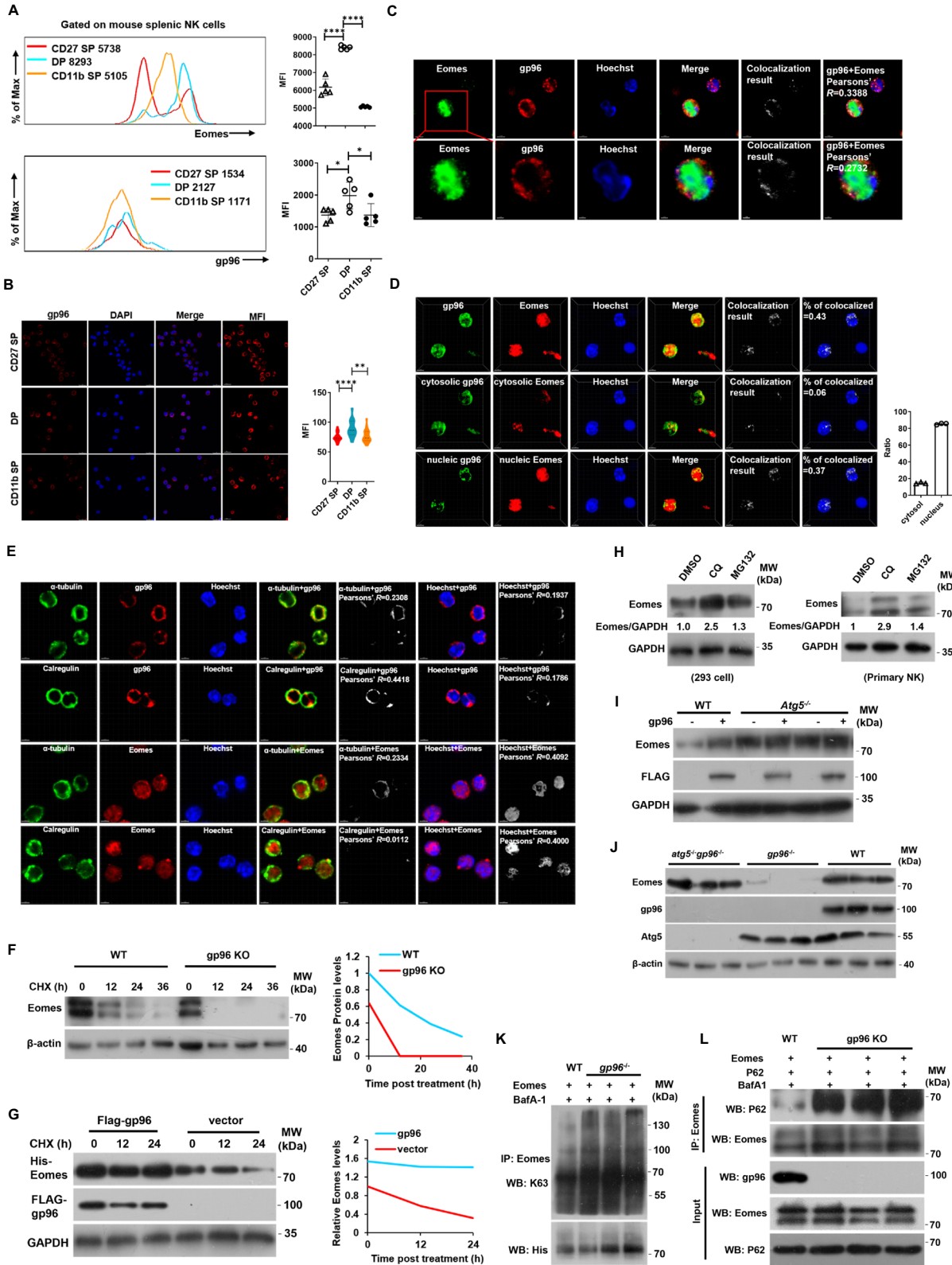

## Methods

### Mice

*Gp96^{fl/fl}* mice were provided by Professor Zihai Li (Ohio University, United States of America)[19]. *Ncr1*^{iCre} mice (Stock No. 110127) were purchased from Biocytogen (Beijing, China). *Gp96^{fl/fl}* mice were crossed with *Ncr1*^{iCre} mice to generate NK-specific gp96-deficient mice (referred to as KO mice); age and sex-matched *gp96^{fl/fl}* mice were used as control (referred to as WT mice). *Ncr1*^{Cre} mice were used to rule out

any effects of the Ncr1-iCre allele on the key readouts. CD45.1+ mice have been described previously[45]. In all experiments, female mice were used. All mice were 8 to 12 weeks old, had a B6 background, and were housed in the specific pathogen-free facility at the Institute of Microbiology, Chinese Academy of Sciences. Euthanasia was performed by the Carbon dioxide ($CO_2$) inhalation or cervical dislocation method. Animal studies were carried out according to the guidelines set forth by the Institute of Microbiology, Chinese Academy of Sciences of

**Fig. 6 | Gp96 inhibits autophagy-lysosome-mediated degradation of Eomes.**
**A** Flow cytometry analysis of Eomes and gp96 levels among CD27 single positive, double positive (DP), CD11b single positive NK cells in mouse spleen. Dots represent data from $n = 5$ mice/group. **B** Immunostaining of gp96 in spleen NK cells sorted from WT mice. The mean fluorescence intensity of gp96 was analyzed by Imaris 9.7. Bar, 10 μm. **C–E** CD11b$^+$CD27$^+$ double positive NK cells were sorted from WT mice and subsequently stained for nucleus (Hoechst), Eomes, gp96, Calregulin and α-tubulin for confocal microscopy analysis. Scale bars, 3 μm for original shots (**C–E**) and 1 μm for magnified views (**C**). Z-stack images of gp96 and Eomes in whole cell, cytosol and nucleus, respectively. Percentage of ROI colocalized for cytosol and nucleus were obtained by Imaris. Percentages of colocalization of gp96 and Trim28 in the nucleus and cytosol were calculated, respectively. Scale bars, 3 μm. Dots represent data from $n = 3$ fields. (**D**). **F** Western blot analysis of Eomes levels in spleen NK cells sorted from WT and gp96 KO mice. Cells were treated with 50 μg/ml CHX for the indicated times. The band intensity at 0 h in WT NK cells was arbitrarily taken as 1.0. **G** Western blot analysis of Eomes levels in 293 cells transfected with Flag-gp96 or the empty vector. Cells were treated with 50 μg/ml CHX for the

indicated times. The band intensity at 0 h in vector cells was arbitrarily taken as 1.0. **H** 293 cells stably expressing Eomes (left) and NK cells from gp96 KO mice (right) were treated with either 40 μM CQ or 10 μM MG132 for 6 h and subjected to western blotting. **I** WT and *Atg5* knockout 293 cells were transfected with His-Eomes and Flag-gp96 or a control vector. Cells were lysed and subjected to western blotting. **J** WT, *Atg5* and *gp96/Atg5* double knockout 293 cells were transfected with His-Eomes and subjected to western blotting. **K** WT and gp96 knockout HEK293 cells were transfected with His-Eomes. Cells were grown for 24 h and treated with 20 nM of BafA1 for 4 h, followed by IP-Western analyses. **L** WT and *gp96* knockout 293 cells were co-transfected with His-Eomes and GFP-P62. Cells were then treated as (**H**) and subjected to IP-Western analyses. The data are representative of two independent experiments with similar results. Mean ± SD is shown. Statistical significance was determined using two-tailed unpaired *t* test. *$p < 0.05$, **$p < 0.01$, ****$p < 0.0001$. *p* values: (A) $p < 0.0001$ (Eomes, CD27 SP vs DP), $p < 0.0001$ (Eomes, DP vs CD11b SP), $p = 0.0193$ (gp96, CD27 SP vs DP), $p = 0.0391$ (gp96, DP vs CD11b SP), (**B**) $p < 0.0001$ (CD27 SP vs DP), $p = 0.0097$ (DP vs CD11b SP).

---

Research Ethics Committee under the approved protocol numbers PZIMCAS2011001.

### Cell lines and plasmid construction
B16F10 cells from our laboratory were cultured in-house. The HEK293 cell line, NK92 human NK cell lines, MC38 colon cancer cells, and LLC lung cancer cells were purchased from the Cell Resource Center, Peking Union Medical College. The cell line was checked free of mycoplasma contamination by PCR and culture. Its species' origin was confirmed with PCR. The cell line's identity was authenticated with STR profiling (FBI, CODIS). All the results can be viewed on the website (http://cellresource.cn). B16F10 cells were cultured in RPMI 1640 medium with 10% fetal bovine serum (FBS), penicillin, and streptomycin (100 IU/ml). HEK293, MC38, and LLC cells were cultured in Dulbecco's modified Eagle medium with 10% FBS, penicillin, and streptomycin (100 IU/ml). NK92 cells were cultured in α-MEM with 12.5% FBS, 12.5% horse serum, penicillin, and streptomycin (100 IU/ml), and IL-2 (10 ng/ml). All plasmids used in this paper were purchased by iGene Biotechnology Co., Ltd.

### Western blot and co-immunoprecipitation assay
The stimulated NK92 cells or sorted splenic NK cells were harvested, and an equal number of cells were directly lysed. Primary and secondary antibodies used in western blot and were listed in Supplementary Table 3. For endogenous Co-IP, cells were lysed with 0.25% NP-40 lysis buffer (20 mM Tris-HCl, 125 mM NaCl, 5 mM MgCl2, 0.2 mM EDTA, 12% Glycerol, and 0.25% Nonidet P-40). Equal amounts of total protein were incubated with indicated antibodies or normal mouse IgG overnight at 4 °C, and then 40 μl of protein A/G beads were added for an additional 2 h of incubation. For exogenous Co-IP, anti-HA beads (or anti-flag beads, or anti-myc beads) were added to equal amounts of total protein and incubated for 3 h. Beads were centrifuged (800 rpm for 2 min) and washed five times using wash buffer (20 mM Tris-HCl, 125 mM NaCl, 5 mM MgCl$_2$, 0.2 mM EDTA, and 0.1% Nonidet P-40). The beads were heated at 100 °C for 10 min before SDS-PAGE and immunoblotting.

### Cell fractionation
Cell fractions were harvested using Plasma Membrane Protein Isolation and Cell Fractionation Kit (Invitrogen) following the manufacturer's instructions. Fourty micrograms of protein were analyzed by SDS-PAGE and western blot.

### Immunostaining for confocal analysis
Murine NK cells were isolated from the spleen by magnetic cell sorting using EasySep Mouse NK Cell Isolation Kit (Stemcell) following the manufacturer's instructions. A total of 50-90% pure NK

cells were obtained using this procedure. Cells were then subsequently sorted into different subsets using a FACSAria Cell Sorter (Becton–Dickinson, San Jose, USA). After 4% PFA fixation for 15 min, blocking (3% BSA in PBS) for 30 min at RT was performed. Immunostainings were performed after a permeabilization step with 0.05% Triton X-100 for 30 min. Primary antibodies were added to the cells for one-hour incubation at RT. After washes with PBS, cells were incubated with the appropriate secondary antibodies along with Hoechst for 30 min at RT. After washes with PBS, cells were observed with a Leica SP8 laser scanning confocal microscope. The images and relative quantification were processed using Imaris (ver 9.7.2).

Z-stack images of gp96 and Trim28 in whole cell, cytosol and nucleus were generated by Imaris. Before analysis, a new surface for nucleus was created using the Hoechst signal. Nucleic signal and cytosol signal were obtained by the mask channel function in Imaris. Signals retained in the cell nucleus are considered as nuclear gp96 or Trim28, while signals outside the nucleus are regarded as cytosolic gp96 or Trim28. The resulting images were processed by analyzing colocalization, and the percentage of ROI colocalized for cytosol and nucleus were obtained respectively. A total of 3 fields randomly were selected and quantified in each assay. The 3D videos were provided in the supplementary materials. Primary and secondary antibodies used in immunofluorescence were listed in Supplementary Table 3.

### In vivo ubiquitination assays
For the in vivo ubiquitination assay, HEK293 cells were co-transfected with expressing plasmids encoding Myc-Eomes and Trim28. Cells were grown overnight and treated with 20 nM Bafilomycin A1 (BafA1) for 4 h before harvesting. Cell lysates were immunoprecipitated using anti-Myc resin, followed by Western blot analyses with ubiquitin, ubiquitin-Lys 48-only or ubiquitin-Lys 63-only.

### Flow cytometry
The spleen and LNs were collected and single-cell suspensions of lymphocytes were prepared by mechanical disruption in 1640 medium supplemented with 2% FBS. Peripheral blood mononuclear cells (PBMC) were freshly isolated from healthy donors. Written informed consent was obtained for each participant according to institutional guidelines. All experiments were approved by the Fifth Medical Center of PLA General Hospital (permit number KY-2020-11-5-1). Single-cell suspensions were stained with the appropriate monoclonal antibody in phosphate-buffered saline containing 5% serum. To detect phosphorylated signaling proteins, NK cells were fixed with Phosflow Lyse/ Fix buffer, followed by permeabilization with Phosflow Perm buffer III (BD) and staining with antibodies. All other intracellular proteins were stained according to the manufacturer's instructions using Foxp3/ Transcription Factor Staining Buffer Set Kit (eBioscience). Fortessa and

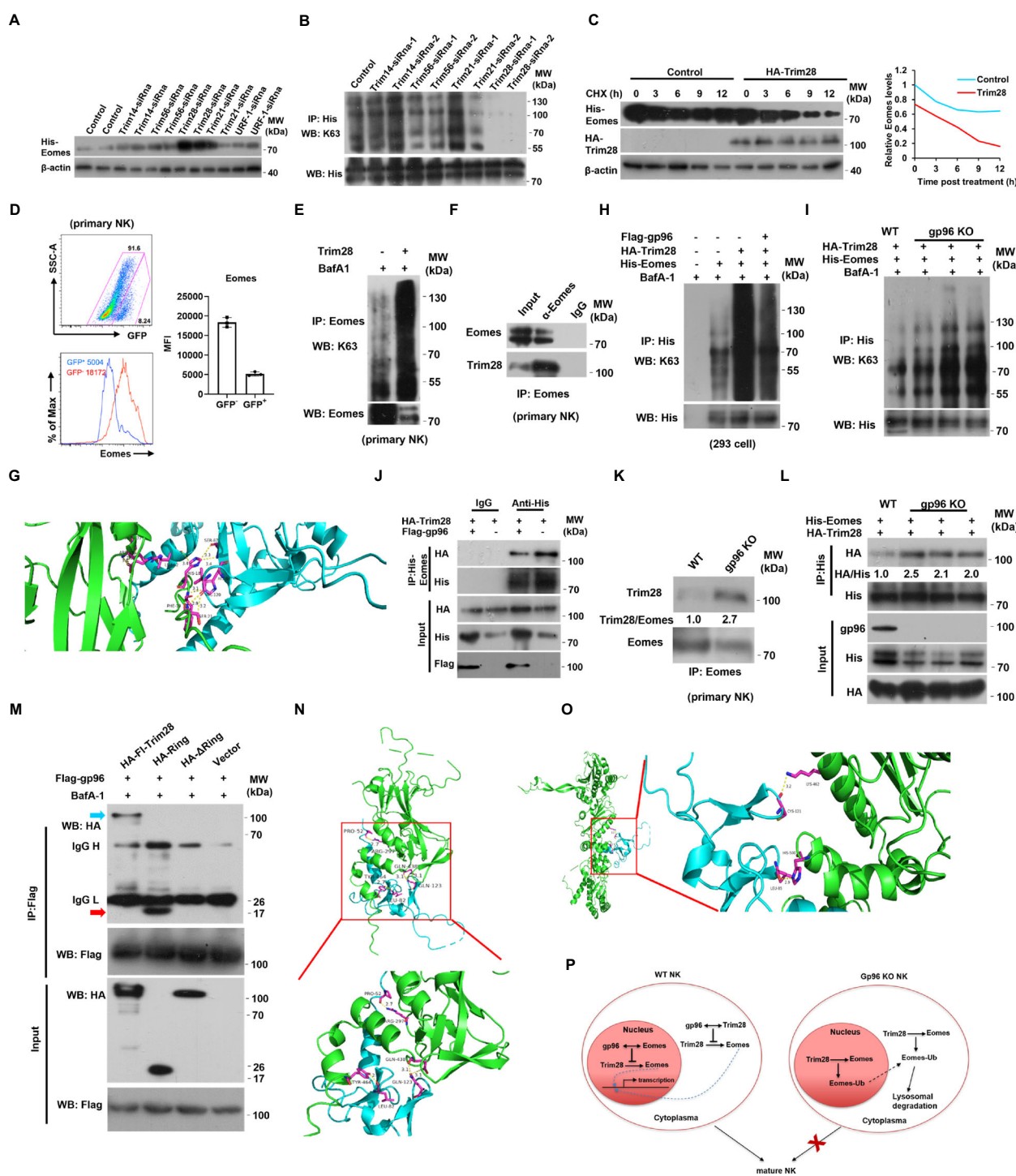

FACSAria III (BD Biosciences, San Diego, USA) were used for analysis and cell sorting, with dead cells excluded by the LIVE/DEAD Fixable Violet Dead Cell Stain Kit (Invitrogen, Carlsbad, USA). Primary antibodies used in flow cytometry were listed in Supplementary Table 4.

### RNA-seq and analysis
CD3⁻NK1.1⁺DX5⁺ NK cells were sorted to a typical purity of >98%. RNA-seq and bioinformatics analysis were conducted by Shanghai Biotechnology Corporation.

### Single-cell RNA sequencing and data processing
Freshly isolated splenocytes were stained with anti-CD3 (17A2), and anti-NK1.1 (PK136) antibodies and NK cells were sorted to >99% purity

using a BD FACSAria™ III. Splenic NK cells pooled from three mice of each indicated genotype were enriched by FACS for library preparation, and scRNA-seq was performed by Shanghai Biotechnology Corporation. The final library pool was sequenced on the Illumina NovaSeq 6000 instrument using 150-base pair paired-end reads. Raw sequencing data were converted to FASTQ files and aligned to the mouse genome reference sequence (GRCH38). The 10X Genomics Cell Ranger (version 3.0.1) was used to demultiplex samples, process barcodes, and generate a digital gene-cell matrix from this data. The output was then imported into the Seurat (version 3.1.5) R toolkit for quality control and downstream analysis. The percentage of ribosome genes is 0.326 and 0.438% in gp96^fl/fl^ and Ncr1^Cre^gp96^fl/fl^ cells, respectively. Cells with less than 200 genes and cells with above 20%

**Fig. 7 | Gp96 blocks Trim28-mediated Eomes degradation by completion with Eomes for Trim28 binding. A, B** HEK293 cells stably expressing Eomes were transfected with siRNAs targeting indicated E3 ligases. Cells were then subjected to western blotting (**A**). Cells were grown for 24 h and treated with 20 nM of BafA1 for 4 h, followed by IP-Western analyses (**B**). **C** HEK293 cells stably expressing Eomes were transfected with Trim28 or a control vector. Cells were treated with 50 µg/ml CHX for the time as indicated and were then subjected to western blotting. **D** Flow cytometry analysis of Eomes levels in GFP⁻ and GFP⁺ cell in primary NK cells infected with Trim28 GFP lentiviral vector (MOI = 10). $n$ = 3 biologically independent samples. Mean ± SD is shown. Primary NK cells transfected with Trim28 were treated with 20 nM of BafA1 for 4 h, followed by IP-Western analyses (**E**), or endogenous Eomes protein was immunoprecipitated with a specific antibody for Eomes or normal rabbit IgG, followed by immunoblotting (**F**). **G** The protein-protein interaction prediction tools-ZDOCK 3.0.2 was used to predict the interaction between full-length Eomes (PDB ID: AF-O54839) and Trim28 (PDB ID: AF-Q62318). Blue represents Trim28, and the green represents Eomes. **H** HEK293 cells transfected with indicated plasmids were grown for 24 h and treated with 20 nM of BafA1 for 4 h, followed by IP-Western analyses. **I** WT and gp96 knockout HEK293 cells were transfected with His-Eomes and HA-Trim28 plasmids. Cells were grown for 24 h and treated with 20 nM of BafA1 for 4 h, followed by IP-Western analyses. **J** HEK293 cells stably expressing Eomes were transfected with indicated plasmids. Cells were grown for 24 h, followed by IP-Western analyses. **K** WT and gp96 knockout HEK293 cells were transfected with His-Eomes and HA-Trim28 plasmids. Cells were grown for 24 h, followed by IP-Western analyses. **L** Primary NK cells from WT and gp96 knockout mice were sorted, and endogenous Eomes protein was immunoprecipitated with a specific antibody for Eomes or normal rabbit IgG, followed by immunoblotting. **M** HEK293 cells stably expressing Flag-gp96 were transfected with indicated plasmids. Cells were grown for 24 h, followed by IP-Western analyses. **N** The protein-protein interaction prediction tools-ZDOCK 3.0.2 were used to predict the interaction between Eomes (PDB ID: AF-O54839) and Trim28 (PDB ID: AF-Q62318). Blue represents the Ring domain of Trim28, and the green represents Eomes. **O** The protein-protein interaction prediction tools-ZDOCK 3.0.2 were used to predict the interaction between gp96 (PDB ID: AF-P14625) and Trim28 (PDB ID: AF-Q62318). Blue represents the Ring domain of Trim28, and the green represents gp96. **P** A working model. The E3 ubiquitin ligase Trim28 targets Eomes for lysosomal degradation, resulting in the inhibition of NK development and function. Gp96 binds to Trim28 mainly in cytosol and Eomes mainly in nucleus, and protects Eomes from Trim28-mediated degradation. The data are representative of two independent experiments with similar results.

mitochondrial genes were removed as low-quality cells. Cells with abnormally high unique molecular identifiers or gene numbers were also identified as putative doublets and removed. After quality control and filtering steps, 7349 cells from *gp96^{fl/fl}* mice and 8350 cells from *Ncr1^{Cre}gp96^{fl/fl}* mice were used for further analyses. The Louvain algorithm was used for the unsupervised computational analysis of scRNA-seq data.

## Generation of BM chimeras

To generate mixed BM chimeras, BM cells were isolated from the femurs and tibias of CD45.1⁺WT and CD45.2⁺ *Ncr1^{Cre}gp96^{fl/fl}* mice. WT recipient mice (CD45.1⁺CD45.2⁺) were sub-lethally gamma-irradiated (7 Gy) and intravenously transplanted with a mixture (1:1) of WT (CD45.1) and KO (*Ncr1^{Cre}gp96^{fl/fl}*) donor bone marrow cells (5 × 10⁶). NK cells were analyzed 8 weeks after reconstitution.

## Cytolytic assays

For cytolytic assays against YAC-1 cells, CFSE−labeled YAC-1 cells were cocultured with effector cells (mouse splenic NK cells) at the indicated effector:target (E:T) ratios for 4 hours. Next, cell mixtures were stained with 7-AAD to calculate killing efficiency as the percentage of 7-AAD⁺ YAC-1 cells with flow cytometry. To determine NK cell cytotoxic activity, 2 × 10⁶ CFSE-labeled YAC-1 cells were intraperitoneally injected into WT and KO mice, respectively. Cells in the peritoneal cavity were harvested, and CFSE-labeled YAC-1 cells were measured 24 h post-injection.

## Transfection of primary NK cells

Murine NK cells were sorted from the spleen using EasySep Mouse NK Cell Isolation Kit (Stemcell) following the manufacturer's instructions. 3 × 10⁶ NK cells were placed into each well of a 12-well culture plate and then transfected using Advanced DNA RNA Transfection Reagent (ZETA LIFE, AD600025). 10 µg plasmid was directly mixed with transfection reagent according to 1:1 relationship, then use a pipette to blow 10-15 times to mix. After incubation at room temperature for 15 minutes, the transfection complex was prepared. Add transfection complex to the cell culture plate and mix gently, place in the CO₂ incubator and continue to culture. Cells were grown for 48 h, followed by IP-Western analyses.

## Isolation of progenitor stem cells and NK differentiation in vitro

Bone marrow cells were obtained from WT or KO mice. Progenitor stem cells (PSCs) were enriched using an EasySep Mouse Progenitor Stem Cell Negative Selection kit (StemCell Technologies, Canada).

Transduction was conducted immediately after the enrichment of PSCs using the spin protocol. PSCs (0.3 × 10⁶) were placed into each well of a 48-well culture plate. After transduction, the virus-containing supernatant was removed. The transduced PSCs were cultured in NK differentiation conditioned medium[46] RPMI 1640 supplemented with 10% FBS, 1% PSG, 1.6 mmol/l 2-mercaptoethanol, 0.5 ng/ml of murine IL-7, 30 ng/ml of stromal cell factor, and 50 ng/ml FMS-related tyrosine kinase 3 ligand. On day 3, after transduction, 0.5 ml of fresh media was added. On day 5, cells were pelleted, old media was removed, and cultured in 0.5 ml of complete RPMI media containing 30 ng/ml of IL-15. On day 8, 0.5 ml of fresh media was added. On day 10, old media was replaced with fresh media containing 30 ng/ml of IL-15 and 1000 U/ml of IL-2. On day 14, differentiated NK cells were analyzed by flow cytometry.

## Virus production and transduction protocols

Bone marrow progenitor cells and primary NK cells were enriched by magnetic cell sorting. The transduction of cells was performed in the following protocol. Cells were pelleted at 400 × $g$ for 5 min at room temperature in 1.5 ml screw-cap tubes and resuspended in 0.25 ml of viral supernatant (MOI = 10) in the presence of 8 µg/ml of Polybrene (Sigma, St. Louis, MO). Cells were incubated for 2 h at 37 °C and 5% CO₂. At the end of transduction, the virus-containing supernatant was removed.

## BM NK differentiation in vitro

Bone marrow cells were cultured in RPMI1640 medium containing 10% fetal calf serum and 100 ng/ml stem-cell factor (SCF), with or without 50 ng/ml rIL15 (Genzyme). Flow cytometric and cytolytic analyses were done after culture for 10 days.

## Tumor models

Single-cell suspension of MC38 colon cancer or LLC lung cancer cells was injected subcutaneously into female WT or KO mice (2 × 10⁵ cells per mouse). Mice were euthanized on days 21 to 28 following tumor injection for analysis of tumor-infiltrating lymphocytes. Tumor volumes were monitored with a caliper and calculated using the formula: V (in mm³) = 0.5 (ab²), where a is the longest diameter and b is the shortest diameter.

All animal experiments were performed in strict accordance with institutional guidelines on the handling of laboratory animals. Mice were euthanized when the maximum tumor size (diameter: 2.0 cm) had been reached. The diameter of the tumor is measured by a caliper. The health of the animals was monitored every other

day and there were no unexpected deaths. To minimized mice suffering and distress, the only tumor was implanted in the subcutaneous site in the flank on each animal. Criteria for early termination are listed as below: If the tumor meets the tumor size limitations; If the tumor becomes ulcerated or necrotic; If the animal's ability to eat or drink is compromised; If the animal's ability to ambulate normally or breathe is impaired. Cervical dislocation was applied in the study to minimized animal suffering and distress.

## Isolation of tumor-infiltrating lymphocytes

Tumor-infiltrating lymphocytes (TILs) were isolated by dissociating tumor tissue in the presence of collagenase I (0.1% w/v, Sigma) and DNase (0.005% w/v, Sigma) for 1 h before centrifugation on a discontinuous Percoll gradient (GE Healthcare). After centrifugation, the white opaque layer at the interface between the two Percoll solutions, mostly containing leukocytes, is collected. Isolated cells were then used in various assays of NK cell function.

## Intravenous lung metastasis assay

For lung experimental-metastasis studies, $2.5 \times 10^5$ B16F10 cells were injected intravenously into WT or KO mice. The number of B16F10 melanoma surface nodules in the lungs of each mouse was counted. Two weeks after the injection, lung tumor nodules were counted.

## Quantification and statistical analysis

FACS data were collected and processed using FACS software (FlowJo, version 10). GraphPad Prism 6 software was used to analyze data using a two-tailed paired Student's *t* test or two-way ANOVA. Data are represented as mean ± SD. $P < 0.05$ were considered significant.

## Reporting summary

Further information on research design is available in the Nature Portfolio Reporting Summary linked to this article.

# Data availability

The RNA-seq data generated in this study have been deposited in the National Center for Biotechnology Information Sequence Read Archive (NCBI SRA) under accession number PRJNA870662. The scRNA-seq data generated in this study have been deposited in the National Center for Biotechnology Information Sequence Read Archive (NCBI SRA) under accession number PRJNA872215. Source data are provided with this paper.

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

## Acknowledgements
We would like to thank Tong Zhao for FACS sorting and technical help. We thank Xiaolan Zhang for performing the confocal laser scanning microscopy and SIM imaging. This work was supported by a grant from National Key R&D Program of China (2022YFC2304203), the Strategic Priority Research Program of the Chinese Academy of Sciences (XDB29040000), the Industrial innovation team grant from Foshan Industrial Technology Research Institute (2018HS), and grants from the National Natural Science Foundation of China (81871297, 81903142 and 32070163), Foshan High-level Hospital construction DengFeng plan, and Guangdong Province biomedical innovation platform construction project tumor immunotherapy.

## Author contributions
S. Meng and X. Li conceived the project. Z. Li and M. Fang supervised the project. X. Li and Y. Xu performed the experiments and analyzed the data. F. Cheng and B. Zhao assisted with the Bioinformatics analysis. X. Li and S. Meng wrote the manuscript.

## Competing interests
The authors declare no competing interests.
