## [Peer Review File · Nature Communications]

REVIEWER COMMENTS

Reviewer #1 (Immune cell development, TF regulation) (Remarks to the Author):

In their study “Heat shock protein gp96 drives natural killer cell maturation and anti-tumor immunity by counteracting Trim28 to stabilize Eomes”, Yuxiu Xu et al. have identified Glucose-regulated Protein 94, also known as GP96 or HSP90B1, as a critical regulator of NK cell maturation. Mice with NKp46-specific deletion of Gp96 are developmentally stalled at the immature CD27+CD11b- stage. Consequently, Gp96-deficient NK cells are defective effector cells with significantly diminished IFN- γ production and cytotoxic activity. As a result, mice with NKp46-specific deletion of Gp96 demonstrate impaired anti-tumor immune responses when challenged with Lewis lung carcinoma (LLC), colon carcinoma (MC38), and melanoma cell lines.

Bone marrow chimera experiments demonstrated that the NK cell maturation defect associated with Gp96-deficiency is cell-intrinsic. Their investigations have revealed that GP96 plays an important role in the IL-15/JAK/STAT5 and mTOR signaling pathways, and in EOMES protein stability. Since ectopic expression of Eomes in HSCs restored NK cell maturation, the authors focused on the role of GP96 in regulating EOMES expression. Their findings reveal a novel function of GP96 in regulating EOMES protein stability by blocking EOMES interaction with E3 ligase TRIM28. In the absence of GP96, EOMES interacted with TRIM28 and was ubiquitinated at lysine 63, targeting it for autophagy-lysosomal-mediated degradation.

The conclusions in this study are supported by the results. The authors have used a multitude of experimental approaches (biochemistry, imaging, RNA-Seq, siRNA perturbation studies) to demonstrate the specificity of GP96/TRIM28 interaction and its consequences on EOMES expression.

Minor comment:

Line 234, I think the authors are referring to Figure 6D and 6E (not Figure 4D and 4E).

Reviewer #2 (Stress response/UPR/HSP) (Remarks to the Author):

The authors demonstrated that very new and important role of gp96 in the maturation and cytotoxic activity of NK cells through the interaction Eomes and Trim 28.

Their findings are very interesting. However, the precise interaction sites (colocalization) among gp96, Eomes and Trim28 are not clear and should be definitively demonstrated.

Major points;

1. Gp96 is an ER-resident HSP. Eomes is a transcription factor and is mainly localized within nucleus. Trim28 is also known to localize within nucleus. Then, which is the site for the interaction within cells among these 3 molecules?
2. In relation to the above, Fig. 6C and D, those are difficult to judge. This reviewer suggests to show magnified views of each photo. Especially, Fig 6D is the key figure to show the colocalization of Eomes and gp96. Moreover, I strongly suggest to show the colocalization of Trim28, gp96, and Eomes using appropriate ER markers.
3. Fig. 6D and 6E, the authors used CD27 single positive NK cells but not Eomes and gp96 highly positive cells, which are double positive (DP, CD16+, CD27+) cells. Is there any reason for choosing this population? I recommend to show the results using DP cells because gp96 is expressed at the high level and it can be clearly shown the colocalization among gp96, Eomes and Trim28.
4. Was the localization of Eomes and Trim28 in gp96 KO NK cells same as WT NK cells? Please show these data.

Minor points;

1. line 243, (Fig. 4D and 4E) is not correct and it should be (Fig. 6D and 6E).
2. In the Methods section, methods for the analysis using confocal microscopy are missing and should be described.

Reviewer #3 (NK biology, scRNAseq) (Remarks to the Author):

In this manuscript, Xu et al demonstrate that NK cell-specific deletion of the heat shock protein, gp96, in mice leads to defects in NK cell differentiation and maturation *in vivo*, including impaired generation of mNKs in general and of highly mature CD27-CD11b+ mNKs in particular. These defects coincided with reduced NK cell responsiveness to IL-15, diminished NK cell effector responses, and increased susceptibility to tumors *in vivo*.

Overall, this is an interesting, thorough, and likely impactful study that reveals a novel role for gp96 in regulating key aspects of NK cell differentiation and function. The work is expected to appeal to a broad audience of immunologists, in addition to NK cell biologists.

The manuscript provides convincing evidence that gp96 is a major regulator of mouse NK cell biology. The work also includes intriguing mechanistic studies aimed at demonstrating that gp96 may regulate NK biology by protecting Eomes from Trim28-mediated ubiquitination and subsequent degradation. While highly interesting, these mechanistic studies are less convincing than other aspects of the study for two main reasons. First, the impact of gp96-deficiency on Eomes expression in NK cells appears to be very modest, at least by flow cytometry (Fig. 5C), and biological/experimental replicates addressing the reproducibility of this effect are not shown (e.g., Fig. 5C-D). Second, the mechanistic studies (Figures 6F-J and 7A-I and 7K) are principally performed using non-NK cell lines (e.g. HEK293), so the extent to which the findings are relevant to NK cells is unclear. Confirmation of the key mechanistic findings in NK cells (preferably primary NK cells), would substantially increase the impact of the work.

In addition, the following minor concerns were also noted:

1. Lines 55-59: please clarify that mature CD27-CD11b+ NK cells also exist in the bone marrow and indicate what is meant by “optimal” and “optimally”.
2. Lines 59-60: it should be noted that additional stages of NK cell maturation have been described, e.g., highly mature KLRG1+ NK cells.
3. Line 61 and throughout: please distinguish between immature NK cells (iNKs) and less mature CD27+CD11b- mNK cells throughout the manuscript
4. For studies depicted in Figure 1C-E, are the frequencies of CD27-, CD11b-, and other receptor-expressing cells also observed in gp96 KO NK cells if the analyses are performed using a CD3-NK1.1+ gate, rather than a CD3-NK1.1+DX5+ gate? If gp96-deficiency alters DX5/CD49b expression, it may be unsuitable for use in an upstream gate.
5. Figure 3C: it is surprising that NK cells could be recovered after 7 days in culture without any IL-15.
6. Line 182: re-phrase “we corresponded”
7. Line 200: define “terminal” or re-phrase
8. Line 211: Please clarify which cell type(s) were used for the referenced ChIPseq studies, and how Eomes KO mice were used to identify Eomes-bound loci.
9. Figure 5A: Clarify whether the y-axis show Log2 fold change of WT vs KO or KO vs WT.
10. Line 237: the word “dramatic” does not seem appropriate; the effect appears to be modest or moderate.

11. Figure 6F/6G: please clarify whether and how Eomes levels were normalized for quantification.

Reviewer #1 (Immune cell development, TF regulation) (Remarks to the Author):

In their study “Heat shock protein gp96 drives natural killer cell maturation and anti-tumor immunity by counteracting Trim28 to stabilize Eomes”, Yuxiu Xu et al. have identified Glucose-regulated Protein 94, also known as GP96 or HSP90B1, as a critical regulator of NK cell maturation. Mice with NKp46-specific deletion of Gp96 are developmentally stalled at the immature CD27+CD11b- stage. Consequently, Gp96-deficient NK cells are defective effector cells with significantly diminished IFN- γ production and cytotoxic activity. As a result, mice with NKp46-specific deletion of Gp96 demonstrate impaired anti-tumor immune responses when challenged with Lewis lung carcinoma (LLC), colon carcinoma (MC38), and melanoma cell lines.

Bone marrow chimera experiments demonstrated that the NK cell maturation defect associated with Gp96-deficiency is cell-intrinsic. Their investigations have revealed that GP96 plays an important role in the IL-15/JAK/STAT5 and mTOR signaling pathways, and in EOMES protein stability. Since ectopic expression of Eomes in HSCs restored NK cell maturation, the authors focused on the role of GP96 in regulating EOMES expression. Their findings reveal a novel function of GP96 in

regulating EOMES protein stability by blocking EOMES interaction with E3 ligase TRIM28. In the absence of GP96, EOMES interacted with TRIM28 and was ubiquitinated at lysine 63, targeting it for autophagy-lysosomal-mediated degradation.

The conclusions in this study are supported by the results. The authors have used a multitude of experimental approaches (biochemistry, imaging, RNA-Seq, siRNA perturbation studies) to demonstrate the specificity of GP96/TRIM28 interaction and its consequences on EOMES expression.

Minor comment:

Line 234, I think the authors are referring to Figure 6D and 6E (not Figure 4D and 4E).

Answer:

As you suggested, we have corrected this error.

Thank you.

Reviewer #2 (Stress response/UPR/HSP) (Remarks to the Author):

The authors demonstrated that very new and important role of gp96 in the maturation and cytotoxic activity of NK cells through the interaction Eomes and Trim 28.

Their findings are very interesting. However, the precise interaction sites (colocalization) among gp96, Eomes and Trim28 are not clear and should be definitively demonstrated.

Major points;

1. Gp96 is an ER-resident HSP. Eomes is a transcription factor and is mainly localized within nucleus. Trim28 is also known to localize within nucleus. Then, which is the site for the interaction within cells among these 3 molecules?

Response: As you suggested, super-resolution immunofluorescence imaging and cell fractionation were performed to discern the interaction site among gp96, Eomes and Trim28 in NK cells. Please see revised Fig.6E, Fig. S6A and S6B. The following sentences have been added in lines 306-313.

“Finally, we determined the interaction sites among gp96, Trim28 and Eomes in NK cells. As can be seen in Fig. 6E, Fig. S6A and S6B, both super-resolution immunofluorescence imaging and analysis of subcellular fractions showed that gp96 is mainly localized in the ER, and a small fraction of gp96 also exists in the cytosol

and nucleus, which was consistent to previous studies^{1,2}. While most of Eomes and Trim28 exist within the nucleus, small amounts of them were found to be distributed in the cytosol. We therefore deduce that the interaction site among these 3 molecules may be the nucleus or even the cytosol.”

Thank you for your comments that greatly strengthened the manuscript.

2. In relation to the above, Fig. 6C and D, those are difficult to judge. This reviewer suggest to show magnified view of each photos. Especially, Fig 6D is the key figure to show the colocalization of Eomes and gp96. Moreover, I strongly suggest to show the colocalization of Trim28, gp96, and Eomes using appropriate ER markers.

Response: As you suggested, colocalization of Eomes and gp96 was determined by using magnified view of photos (zoom factor=4.0 and zoom factor=10.0). In addition, colocalization of Trim28, gp96, or Eomes was analyzed by immuno-colocalization with cytosol marker (α -tubulin), ER marker (calregulin/calreticulin) and nucleic marker (Hoechst), respectively. Images were analyzed and Pearsons' *R* were calculated as indicators of colocalization. Please see revised Fig. 6D and 6E. The following sentence has been added in lines 244-246. “Confocal microscopy results showed that gp96 colocalized with Eomes mainly in the cytosol and nucleus but not in the ER of CD11b⁺CD27⁺ NK cells (Fig. 6D and 6E).”.

Next, we performed immuno-colocalization between Trim28 and gp96, and Eomes and Trim28, respectively. Please see Fig.S6C. The following

sentence has been added in lines 313-314. “The colocalization between Eomes and Trim28, or Trim28 and gp96 was confirmed in CD11b⁺CD27⁺ NK cells (Fig. S6C).”.

Thank you.

3. Fig. 6D and 6E, the authors used CD27 single positive NK cells but not Eomes and gp96 highly positive cells, which are double positive (DP, CD16⁺, CD27⁺) cells. Is there any reason for choosing these population? I recommend to show the results using DP cells because gp96 is expressed at the high level and it can be clearly shown the colocalization among gp96, Eomes and Trim28.

Response: In our previous study, CD27 single positive NK cells were chosen as they were relatively immature compared to DP or CD11b single positive cells. As you suggested, DP cells were sorted by FACS and stained for gp96, Eomes and Trim28. Images were analyzed, and Pearson's *R* was calculated as indicators of colocalization between Trim28 and gp96, Eomes and gp96, and Eomes and Trim28. Please see revised Fig. 6D and 6E, and Fig. S6C. The following sentences have been added in lines 244-246, and lines 313-314. “Confocal microscopy results showed that gp96 colocalized with Eomes mainly in the cytosol and nucleus but not in the ER of CD11b⁺CD27⁺ NK cells (Fig. 6D and 6E).”; “The colocalization between

Eomes and Trim28, or Trim28 and gp96 was confirmed in CD11b⁺CD27⁺ NK cells (Fig.S6C).”.

Thank you for your suggestions that greatly strengthened the manuscript.

4. Was the localization of Eomes and Trim28 in gp96 KO NK cells same as WT NK cells? Please show these data.

Response: As you suggested, localization of Eomes and Trim28 in gp96 KO NK cells as well as WT NK cells was analyzed by IF immunostaining. Please see Fig. S6E. The following sentence has been added in lines 314-318. “In addition, gp96 KO had no obvious impact on Trim28 localization, but resulted in increased localization of Eomes in the cytosol, which may be due to enhanced Eomes ubiquitination by gp96 KO and its subsequent translocation to the cytosol for autophagy-lysosome-mediated degradation (Fig. S6D).”.

Thank you.

Minor points;

1. line 243, (Fig. 4D and 4E) is not correct and it should be (Fig. 6D and 6E).

Response: As you suggested, we have corrected this error.

Thank you.

2. In the Methods section, methods for the analysis using confocal microscopy are missing and should be described.

Response: As you suggested, methods for confocal microscopy analysis and cell fractionation have been added. Please see lines 432-449.

“Cell fractionation

Cell fractions were harvested using Plasma Membrane Protein Isolation and Cell Fractionation Kit (Invitrogen) following the manufacturer's instructions. Fourty micrograms of protein were analyzed by SDS-PAGE and western blot.

Immunostaining for confocal analysis

Murine NK cells were isolated from the spleen by magnetic cell sorting using EasySep Mouse NK Cell Isolation Kit (Stemcell) following the manufacturer's instructions. A total of 50-90% pure NK cells were obtained using this procedure. Cells were then subsequently sorted into different subsets using a FACSAria Cell Sorter (Becton–Dickinson, San Jose, USA). After 4% PFA fixation for 15 min, blocking (3% BSA in PBS) for 30 min at RT was performed. Immunostainings were performed after a permeabilization step with 0.05% Triton X-100 for 30 min. Primary antibodies were added to the cells for one-hour incubation at RT. After washes with PBS, cells were incubated with the appropriate secondary antibodies along with Hoechst for 30 min at RT. After washes with PBS, cells were observed with a Leica SP8 laser scanning confocal microscope. The images and relative quantification were processed using Imaris 9.7.2.”

Thank you.

Reviewer #3 (NK biology, scRNAseq) (Remarks to the Author):

In this manuscript, Xu et al demonstrate that NK cell-specific deletion of the heat shock protein, gp96, in mice leads to defects in NK cell

differentiation and maturation *in vivo*, including impaired generation of mNKs in general and of highly mature CD27-CD11b⁺ mNKs in particular. These defects coincided with reduced NK cell responsiveness to IL-15, diminished NK cell effector responses, and increased susceptibility to tumors *in vivo*.

Overall, this is an interesting, thorough, and likely impactful study that reveals a novel role for gp96 in regulating key aspects of NK cell differentiation and function. The work is expected to appeal to a broad audience of immunologists, in addition to NK cell biologists.

The manuscript provides convincing evidence that gp96 is a major regulator of mouse NK cell biology. The work also includes intriguing mechanistic studies aimed at demonstrating that gp96 may regulate NK biology by protecting Eomes from Trim28-mediated ubiquitination and subsequent degradation. While highly interesting, these mechanistic studies are less convincing than other aspects of the study for two main reasons. First, the impact of gp96-deficiency on Eomes expression in NK cells appears to be very modest, at least by flow cytometry (Fig. 5C), and biological/experimental replicates addressing the reproducibility of this effect are not shown (e.g., Fig. 5C-D). Second, the mechanistic studies (Figures 6F-J and 7A-I and 7K) are principally performed using non-NK cell lines (e.g. HEK293), so the extent to which the findings are relevant

to NK cells is unclear. Confirmation of the key mechanistic findings in NK cells (preferably primary NK cells), would substantially increase the impact of the work.

Response: As you suggested, experiment replicates have been shown in revised Fig.3F, Fig.5C and 5D, and quantification of the IB bands of p-S6 and Eomes is shown in Fig.3F and Fig. 5D. The following sentence has been added in in Fig. 5 legend in line 856. “Mean \pm SD of three replicates is shown.”

In addition, we confirmed the key mechanistic findings in primary NK cells. Please see Fig.6H, Fig.7D and Fig.7E.” The following sentences have been added in lines 251-253, and lines 272-274. “Expression of Eomes was mostly increased in the presence of a lysosome inhibitor CQ, but not proteasome inhibitor MG132 in both primary NK and HEK293 cells (Fig. 6H), suggesting lysosome mediated degradation of Eomes.”; “Moreover, overexpression of Trim28 in primary NK cells led to reduced Eomes levels (Fig. 7D) and increased Lys 63-linked-polyubiquitin (Fig. 7E).”

Thank you for your comments that greatly strengthened the manuscript.

In addition, the following minor concerns were also noted:

1. Lines 55-59: please clarify that mature CD27-CD11b⁺ NK cells also exist in the bone marrow and indicate what is meant by “optimal” and “optimally”.

Response: As you suggested, we have modified these descriptions of

optimal” and “optimally”, and clarified that mature CD27⁻CD11b⁺ NK cells also exist in the bone marrow. Please see lines 54-58, and lines 62-63. “They develop and start the process of maturation in the bone marrow and reach certain functional status when they migrate to the peripheral and differentiate into the mature status. The minimal cell population size and effector functions at the single cell level are required to launch an efficient anti-tumor NK cell response ³.”; “which then differentiate into CD11b⁺CD27⁻ (CD27⁻) mature NK cells that also exist in the bone marrow ⁴.”

Thank you.

2. Lines 59-60: it should be noted that additional stages of NK cell maturation have been described, e.g., highly mature KLRG1⁺ NK cells.

Response: As you suggested, the following sentence has been added in lines 63-64. “In addition, KLRG1⁺ NK cells are at a highly mature stage of maturation⁵.”

Thank you.

3. Line 61 and throughout: please distinguish between immature NK cells (iNKs) and less mature CD27⁺CD11b⁻ mNK cells throughout the manuscript.

Response: As you suggested, we have distinguished between immature NK cells (iNKs) and less mature CD27⁺CD11b⁻ mNK cells throughout the manuscript. Immature NK cells refer to DX5⁻ NK cells, and less

mature NK cells refer to DX5⁺CD27⁺CD11b⁻ NKs. Please see line 61, line 120, line 123, etc.

Thanks for your suggestion.

4. For studies depicted in Figure 1C-E, are the frequencies of CD27⁻, CD11b⁻, and other receptor-expressing cells also observed in gp96 KO NK cells if the analyses are performed using a CD3⁻NK1.1⁺ gate, rather than a CD3⁻NK1.1⁺DX5⁺ gate? If gp96-deficiency alters DX5/CD49b expression, it may be unsuitable for use in an upstream gate.

Response: As you suggested, we analyzed the frequencies of CD27⁻, CD11b⁻ cells and expression levels of DX5 using a CD3⁻NK1.1⁺ gate. Please see Fig. S1C. The following sentence has been added in lines 125-127. “Similar results were obtained between the analyses using CD3⁻NK1.1⁺ gate (Fig. S1C) and CD3⁻NK1.1⁺DX5⁺gate, so DX5⁺ was used in an upstream gate.”

We thank you for your suggestions which greatly enhanced the quality of our work.

5. Figure 3C: it is surprising that NK cells could be recovered after 7 days in culture without any IL-15.

Response: In Fig. 3C, 1×10^6 splenocytes isolated from WT and KO mice were stimulated with the indicated doses (0, 5, 10, and 20 ng/mL,

respectively) of IL-15 for 7 d. Compared to treatment with IL-15, the proportion of total NK cells was quite lower after 7 days in culture without IL-15. As seen in Fig. 3C, small amounts of DX5⁺ NK cells were detected. by FACS. We speculate this may be due to stimulation of trace amounts of autocrine IL-15 by splenocytes, as 0.75 ng/ml of IL-15 was detected in the supernatants after 7 days of culture of splenocytes with no IL-15 treatment⁶. The following sentence has been added in lines 165-167. “Low proportion of NK cells was detected from splenocytes even after 7 days of culture without IL-15 (Fig. 3C), which may be due to stimulation of trace amounts of autocrine IL-15 and other cytokines by splenocytes⁶.”

Thanks for your suggestion.

6. Line 182: re-phrase “we corresponded”

Response: As you suggested, we have re-phrased “we corresponded” to “we defined”. Please see line 188.

Thank you.

7. Line 200: define “terminal” or re-phrase

Response: As you suggested, we have replaced “terminal” with “late stage”. Please see line 206.

Thank you.

8. Line 211: Please clarify which cell type(s) were used for the referenced ChIPseq studies, and how Eomes KO mice were used to identify Eomes-bound loci.

Response: As you suggested, the following sentences have been added in lines 216-225. “Using the mouse model expressing endogenously tagged EOMES⁷, splenic NKs were isolated for ChIP analysis. WT and EOMES^{-/-} NK cells⁷ were performed for RNA-seq to identify differentially expressed genes. Combination the ChIP-seq analysis with RNA-seq results was used to identify direct EOMES target genes. The overlap between EOMES-bound (ChIP-seq) and EOMES-regulated genes (RNA-seq) was identified as direct EOMES target genes. Analysis of the Eomes target genes revealed that the majority of upregulated transcripts in Eomes knockout NK cells were also upregulated in gp96 depleted NK cells, and most of the downregulated transcripts were downregulated simultaneously (Fig. 5A).”

Thank you for your suggestions which greatly enhanced the quality of our work.

9. Figure 5A: Clarify whether the y-axis show Log2 fold change of WT vs KO or KO vs WT.

Response: As you suggested, we have clarified that the y-axis shows Log2 fold changes of WT vs KO in Figure legends. Please see line 849.

“Y-axis showed Log2 fold changes of WT vs. KO mice.”

Thank you.

10. Line 237: the word “dramatic” does not seem appropriate; the effect appears to be modest or moderate.

Response: As you suggested, we have corrected “dramatic” to “moderate”. Please see line 249. Thank you.

11. Figure 6F/6G: please clarify whether and how Eomes levels were normalized for quantification.

Response: As you suggested, we have clarified how Eomes levels were normalized for quantification in Fig. 6 legends. The bands of Eomes on western blot analysis were scanned by densitometry using Quantity One (Bio-rad). In Fig.6F, the band intensity at 0 h in WT NK cells was arbitrarily taken as 1.0. In Fig.6G, the band intensity at 0 h in vector cells was arbitrarily taken as 1.0. Please see lines 869-870 and line 872. “The band intensity at 0 h in WT NK cells was arbitrarily taken as 1.0.” “The band intensity at 0 h in vector cells was arbitrarily taken as 1.0.”

Thank you.

References

1. Wu, B. *et al.* Heat shock protein gp96 decreases p53 stability by regulating Mdm2 E3 ligase activity in liver cancer. *Cancer Lett.* **359**, 325-334 (2015).
2. Guo, W.C. *et al.* Expression and its clinical significance of heat shock protein gp96 in human osteosarcoma. *Neoplasma* **57**, 62-67 (2010).
3. Bi, J. & Wang, X. Molecular Regulation of NK Cell Maturation. *Front. Immunol.* **11**, 1945 (2020).
4. Kim, S. *et al.* In vivo developmental stages in murine natural killer cell maturation. *Nat. Immunol.* **3**, 523-528 (2002).

5. Huntington, N.D. *et al.* NK cell maturation and peripheral homeostasis is associated with KLRG1 up-regulation. *J. Immunol.* **178**, 4764-4770 (2007).
6. Kadowaki, T. *et al.* Galectin-9 prolongs the survival of septic mice by expanding Tim-3-expressing natural killer T cells and PDCA-1+ CD11c+ macrophages. *Crit. Care* **17**, R284 (2013).
7. Zhang, J. *et al.* Sequential actions of EOMES and T-BET promote stepwise maturation of natural killer cells. *Nat Commun* **12**, 5446 (2021).

REVIEWER COMMENTS

Reviewer #1 (Remarks to the Author):

In their study “Heat shock protein gp96 drives natural killer cell maturation and anti-tumor immunity by counteracting Trim28 to stabilize Eomes”, Yuxiu Xu et al. have identified Glucose-regulated Protein 94, also known as GP96 or HSP90B1, as a critical regulator of NK cell maturation. Mice with NKp46-specific deletion of Gp96 are developmentally stalled at the immature CD27+CD11b- stage. Consequently, Gp96-deficient NK cells are defective effector cells with significantly diminished IFN- γ production and cytotoxic activity. As a result, mice with NKp46-specific deletion of Gp96 demonstrate impaired anti-tumor immune responses when challenged with Lewis lung carcinoma (LLC), colon carcinoma (MC38), and melanoma cell lines.

Bone marrow chimera experiments demonstrated that the NK cell maturation defect associated with Gp96-deficiency is cell-intrinsic. Their investigations have revealed that GP96 plays an important role in the IL-15/JAK/STAT5 and mTOR signaling pathways, and in EOMES protein stability. Since ectopic expression of Eomes in HSCs restored NK cell maturation, the authors focused on the role of GP96 in regulating EOMES expression. Their findings reveal a novel function of GP96 in regulating EOMES protein stability by blocking EOMES interaction with E3 ligase TRIM28. In the absence of GP96, EOMES interacted with TRIM28 and was ubiquitinated at lysine 63, targeting it for autophagy-lysosomal-mediated degradation.

The authors have addressed my concerns.

Reviewer #2 (Remarks to the Author):

The authors showed some results in response to my comments, however, these are not enough.

To show the interaction site correctly is the most important to understand the the role of gp96, Trim28 and Eomes.

1. In Fig. 6E and S6A and B, to demonstrate the interaction between gp96 and Trim28 within nucleus and/or cytosol, I recommend to show the gp96 and Trim28 interaction by immunoprecipitation assay using NK cell extract of the nucleus and cytosol.

2. Moreover, in Fig. 6E, the authors should show the direct colocalization of gp96 and Trim28 (in addition to the usage of organelle markers) in nucleus and cytosol and also percentages of colocalization of gp96 and Trim28 in the nucleus and cytosol, respectively.

3. As described above, I strongly suggest to conclude the interaction sites of gp96 and Trim28 within NK cells because the authors described that line 313, interaction site among 3 molecules may be the nucleus or even the cytosol. In the present for, these description seems too rough.

4. Since the authors described that gp96 bind Trim28 in the cytosol at very low level (in line 311), it is considered that the most of Trim28 does not bind to gp96. In this situation, the most of Trim28 binds to Eomes within nucleus and Eomes should be undergo the degradation. Or can very small amount of complex formation between gp96 and Trim28 in the cytosol regulate Eomes degradation? Please explain the authors opinion.

5. To understand spatial interaction between gp96, Trim28 and Eomes, I strongly suggest to demonstrate the colocalization of these 3 molecules and define the interaction sites using images and also immunoprecipitation assay using cell extract of cytoplasm, nucleus and ER.

Reviewer #3 (Remarks to the Author):

While the authors addressed some concerns raised during the initial review, several important issues remain in the revised manuscript:

1. First, the manuscript contains contradictory data related to the interpretation that loss of gp96 results broadly reduces mNK numbers, but increases (spleen) or does not change (BM) iNKs numbers. Specifically, the data in Figure 1B-C and 3A-C seem to indicate that gp96 NK-KO mice have fewer CD3-NK1.1+DX5+ 'mNKs' (particularly DP and CD27- mNKs) in the BM and spleen, but more CD3-NK1.1+DX5- 'iNKs' in the spleen. However, the scRNAseq data in Figure 4 and Supplemental Figure 2B reveal few if any discernible differences in the relative frequency of less mature vs. more mature NK cells from WT vs gp96 NK-KO mice. Instead, the major difference between the WT and gp96 NK-KO mice appears to be related to Cluster 3 (dominated by KO) and Cluster 4 (dominated by WT). However, careful inspection of the gene signatures for Clusters 3 and 4 suggest that both contain relatively mature NK cells (e.g. high expression of *Itgam*, *Klrg1*, *Gzmb*, *Prf1*, *Cma1*; low expression of *Cd27*, *Xcl1*), and it is unclear why both clusters were not classified as 'Stage 3'. The internal inconsistencies between data in Figs. 1/3 and Figs. 4/S2B raise serious concerns. Are the data in Figs. 1&3 an artifact of using DX5 in the upstream gating

(this possibility is not rigorously ruled out by Fig. S1C)? Are ILC1s being inadvertently classified as iNKs? Is this an artifact of the Ncr1-iCre mouse strain used in this study, which inexplicably appears to lack all expression of NKp46 (Fig. 1D)? With respect to the latter question, it would be prudent to include Ncr1-iCre^{+/-} x gp96^{wt/wt} mice as controls to rule out any effects of the Ncr1-iCre allele on the key readouts. This is particularly important since the Ncr1-iCre mouse strain used in this paper is not the same Ncr1-iCre mouse strain used in most papers in the field (Narni-mancinelli, PNAS, 2011).

2. Second, the flow cytometry data in Fig. 5C, which serve as key evidence that gp96-deficient NK cells have reduced Eomes expression, are wholly unconvincing. Indeed, comparison of Eomes expression data in Figs. 5C vs. 6A, suggest that Eomes staining did not work well, if at all, in the experiment depicted in 5C.

Reviewer #2 (Remarks to the Author):

The authors showed some results in response to my comments, however, these are not enough.

To show the interaction site correctly is the most important to understand the the role of gp96, Trim28 and Eomes.

1. In Fig. 6E and S6A and B, to demonstrate the interaction between gp96 and Trim28 within nucleus and/or cytosol, I recommend to show the gp96 and Trim28 interaction by immunoprecipitation assay using NK cell extract of the nucleus and cytosol.

Response: As you suggested, cell fractionation and immunoprecipitation was performed in NK cells. Please see revised Fig. S6E. The following sentences have been added in lines 325-328. "Interaction between gp96, Eomes and Trim28 was further analyzed by immunoprecipitation from cytosol, nucleus and total membrane (including the ER and plasma membrane). As can be seen in Fig.S6E, interaction between gp96 and Trim28 was detected primarily in cell cytosol."

Thank you.

2. Moreover, in Fig. 6E, the authors should show the direct colocalization of gp96 and Trim28 (in addition to the usage of organelle markers) in nucleus and cytosol and also percentages of colocalization of gp96 and Trim28 in the nucleus and cytosol, respectively.

Response: As you suggested, direct colocalization of gp96 and Trim28 in nucleus and cytosol was analyzed using Imaris 9.7. Using the mask function of Imaris, an approximate 84 % of colocalization of gp96 and Trim28 was observed in cytosol as compared to 16 % in nucleus. Please see revised Fig. S6D. The following sentence has been added in lines 323-325 of the Results section. "Colocalization assays showed that an approximate 84 % of colocalization of gp96 and Trim28 was observed in

cytosol as compared to 16 % in nucleus using the mask function of Imaris (Fig.S6D).”. And the following sentences have been added in lines 497-502 of Methods section. “Before quantification, a new surface for nucleus was created using the Hoechst signal. Nucleic signal and cytosol signal were obtained by the mask channel function in Imaris. The resulting images were processed by analyzing colocalization, and the percentage of ROI colocalized for cytosol and nucleus were obtained respectively. A total of 3 fields randomly were selected and quantified in each assay. The 3D videos were provided in the supplementary materials.”

Thank you.

3. As described above, I strongly suggest to conclude the interaction sites of gp96 and Trim28 within NK cells because the authors described that line 313, interaction site among 3 molecules may be the nucleus or even the cytosol. In the present for, these description seems too rough.

Response: As you suggested, we have now concluded that the interaction site of gp96 and Trim28 within NK cells is primarily cell cytosol, as evidenced by cell fractionation and immunoprecipitation analysis and colocalization assays. Please see lines 323-328. “Colocalization assays showed that an approximate 84 % of colocalization of gp96 and Trim28 was observed in cytosol as compared to 16 % in nucleus using the mask function of Imaris (Fig.S6D). Interaction between gp96, Eomes and Trim28 was further analyzed by immunoprecipitation from cytosol, nucleus and total membrane (including the ER and plasma membrane). As can be seen in Fig.S6E, interaction between gp96 and Trim28 was detected primarily in cell cytosol.”

Thank you.

4. Since the authors described that gp96 bind Trim28 in the cytosol at very low level (in line 311), it is considered that the most of Trim28 does not bind to gp96. In this situation, the most of Trim28 binds to Eomes within nucleus and Eomes should be undergo the degradation. Or can very small amount of complex formation between

gp96 and Trim28 in the cytosol regulate Eomes degradation? Please explain the authors opinion.

Response: As shown in Fig. 6E, Fig. S6A and S6B, a small fraction of gp96 exists in the cytosol and nucleus, and small amounts of Trim28 were found to be distributed in the cytosol. However, immunoprecipitation assays with different NK cell fractions showed that interaction between gp96 and Trim28 was detected primarily in cell cytosol (Fig.S6E). Moreover, the interaction between Eomes and gp96 was detected primarily in cell nucleus (Fig.S6E), which may sterically hinder the binding of Trim28 to Eomes. The following sentences have been added in lines 323-334. “Colocalization assays showed that an approximate 84 % of colocalization of gp96 and Trim28 was observed in cytosol as compared to 16 % in nucleus using the mask function of Imaris (Fig.S6D). Interaction between gp96, Eomes and Trim28 was further analyzed by immunoprecipitation from cytosol, nucleus and total membrane (including the ER and plasma membrane). As can be seen in Fig.S6E, interaction between gp96 and Trim28 was detected primarily in cell cytosol. The interaction between Eomes and gp96 (Fig.S6E) or Trim28 (Fig.S6F) was detected primarily in cell nucleus. ZDOCK analysis revealed that the binding site of gp96 in Eomes is at the interface (Arg296, Arg297, Arg284, Ser303, Ser370, Asn459, Tyr464) of the cleft formed by three helices, and this partly overlaps with the binding site (Arg297, Gln438, Tyr464) of Trim28 (Fig. S6G). This indicates that interaction of gp96 with Eomes may sterically hinder the binding of Trim28 to Eomes.”

Although Eomes and Trim28 mainly exist within the nucleus of NK cells, immunoprecipitation analysis with different cell fractions indicated that not all of the Trim28 binds to Eomes (Fig. S6F). To clarify the role of gp96 on regulation of Trim28-mediated Eomes ubiquitination and degradation, the following sentences have been added in lines 405-414. “Moreover, in cell cytosol gp96 was found to primarily bind to Trim28 (Fig.S6D and S6E), and could compete for Eomes binding to Trim28, thereby protecting Eomes from Trim28-mediated degradation. While in cell nucleus, gp96 dominantly associates with Eomes (Fig.6D and S6E), and may sterically block the interaction of Eomes and Trim28 (Fig.S6G). Indeed, increased cytosolic Eomes

was observed in gp96 KO NK cells, which may be due to increased Eomes ubiquitination by Trim28 and subsequent translocation to the cytosol. Our results provide new insight into understanding the novel regulatory mechanism of Trim28-mediated Eomes ubiquitination that was simultaneously affected by gp96 (Fig.7P).”

Thank you for your comments that greatly strengthened this manuscript.

5. To understand spatial interaction between gp96, Trim28 and Eomes, I strongly suggest to demonstrate the colocalization of these 3 molecules and define the interaction sites using images and also immunoprecipitation assay using cell extract of cytoplasm, nucleus and ER.

Response: As you suggested, we have performed colocalization and immunoprecipitation assays. Please see revised Fig. 6D, and Fig. S6D-S6G. The following sentences have been added in lines 252-254 and in lines 323-334. “Confocal microscopy results showed that gp96 colocalized with Eomes dominantly in the nucleus and minorly in cytosol but not in the ER of CD11b⁺CD27⁺ NK cells (Fig. 6C-6E).”; “Colocalization assays showed that an approximate 84 % of colocalization of gp96 and Trim28 was observed in cytosol as compared to 16 % in nucleus using the mask function of Imaris (Fig.S6D). Interaction between gp96, Eomes and Trim28 was further analyzed by immunoprecipitation from cytosol, nucleus and total membrane (including the ER and plasma membrane). As can be seen in Fig.S6E, interaction between gp96 and Trim28 was detected primarily in cell cytosol. The interaction between Eomes and gp96 (Fig.S6E) or Trim28 (Fig.S6F) was detected primarily in cell nucleus. ZDOCK analysis revealed that the binding site of gp96 in Eomes is at the interface (Arg296, Arg297, Arg284, Ser303, Ser370, Asn459, Tyr464) of the cleft formed by three helices, and this partly overlaps with the binding site (Arg297, Gln438, Tyr464) of Trim28 (Fig. S6G). This indicates that interaction of gp96 with Eomes may sterically hinder the binding of Trim28 to Eomes.”.

Thank you for your suggestions which greatly strengthened our paper.

Reviewer #3 (Remarks to the Author):

While the authors addressed some concerns raised during the initial review, several important issues remain in the revised manuscript:

1. First, the manuscript contains contradictory data related to the interpretation that loss of gp96 results broadly reduces mNK numbers, but increases (spleen) or does not change (BM) iNKs numbers. Specifically, the data in Figure 1B-C and 3A-C seem to indicate that gp96 NK-KO mice have fewer CD3-NK1.1+DX5+ 'mNKs' (particularly DP and CD27- mNKs) in the BM and spleen, but more CD3-NK1.1+DX5- 'iNKs' in the spleen. However, the scRNAseq data in Figure 4 and Supplemental Figure 2B reveal few if any discernible differences in the relative frequency of less mature vs. more mature NK cells from WT vs gp96 NK-KO mice. Instead, the major difference between the WT and gp96 NK-KO mice appears to be related to Cluster 3 (dominated by KO) and Cluster 4 (dominated by WT). However, careful inspection of the gene signatures for Clusters 3 and 4 suggest that both contain relatively mature NK cells (e.g. high expression of *Itgam*, *Klrg1*, *Gzmb*, *Prf1*, *Cma1*; low expression of *Cd27*, *Xcl1*), and it is unclear why both clusters were not classified as 'Stage 3'. The internal inconsistencies between data in Figs. 1/3 and Figs. 4/S2B raise serious concerns. Are the data in Figs. 1&3 an artifact of using DX5 in the upstream gating (this possibility is not rigorously ruled out by Fig. S1C)? Are ILC1s being inadvertently classified as iNKs? Is this an artifact of the *Ncr1-iCre* mouse strain used in this study, which inexplicably appears to lack all expression of NKp46 (Fig. 1D)? With respect to the latter question, it would be prudent to include *Ncr1-iCre*^{+/-} x gp96wt/wt mice as controls to rule out any effects of the *Ncr1-iCre* allele on the key readouts. This is particularly important since the *Ncr1-iCre* mouse strain used in this paper is not the same *Ncr1-iCre* mouse strain used in most papers in the field (Narni-mancinelli, PNAS, 2011).

Response:

As you pointed out, there were inconsistencies between the results of flow cytometry analysis and single-cell RNA-seq of NK cells. We think this may be partly due to different hallmarks used to define these three distinct subsets with NK cell maturation. In flow cytometry analysis, CD27 and CD11b were mainly used to define less mature CD11b⁻CD27⁺ (CD27 SP), CD27⁺CD11b⁺ (DP) and mature CD11b⁺CD27⁻ (CD11b SP) NK subsets on gated NK1.1⁺DX5⁺ cells. While for single-cell RNA-seq, six distinct clusters were defined by hierarchical cluster analysis of transcript signatures (e.g Cma1, Klrg1, Cd7, Xcl1, etc), and NK cell maturation stages were defined by Cd7, Cd27, Cma1, Gzmb, Itgam, Klrg1, Prf1, Xcl1.

To resolve the discrepancy, a subset of CD11b⁻CD27⁻ double negative NK cells were analyzed in flow cytometry analysis. Please see the revised Fig. 1B and 1C, revised Fig. S1C. The following sentences have been added or revised in lines 123-128. “We observed that a subset of immature CD11b⁻CD27⁻(double negative, DN) cells in DX5⁺ NK cells apparently emerged in gp96-deficient mice. Compared to WT mice, gp96-deficient mice contained significantly more immature CD11b⁻CD27⁻ and less mature CD11b⁻CD27⁺ NK cells, and fewer CD27⁺CD11b⁺ and CD11b⁺CD27⁻ mature NK cells in spleen (Fig. 1B). Similar results were observed for NK cells in the bone marrow (Fig. 1C).”

Meanwhile, in scRNA-seq analysis cluster patterns and maturation markers between NK cells from *gp96*^{-/-} and WT mice were compared, especially for cluster 3 and 4. Please see revised Fig.2D. The following sentences have been added or revised in lines 202-220. “These clusters displayed key features in gene expression changes along NK cell maturation, including expressions of genes for effector molecules (Gzmb and Prf1) and genes for maturation markers (CD11b/Itgam, Cma1 and Klrg1) and immature markers (Cd7, and Xcl1)³² (Fig. 4D). Compared to cluster 3, cluster 4 exhibited higher expressions of Itgam, Klrg1, Cma1 and Cd27.

Next, we traced cell fate and reconstructed cell lineage direction using the RNA velocity approach. NK maturation in WT cells followed a single main branch (cluster

3/5 through 2 to cluster 0/1 and 4) without significant division. By contrast, velocity analysis revealed a different branch via cluster 3 through 4 to cluster 0/1 under gp96 knocking out condition (Fig. 4E), which is speculated that cluster 3 may be blocked in a relatively immature developmental stage. Since cluster 3 and cluster 4 are the different major clusters between WT and gp96 KO cells in both cell number and maturation process (Fig. 4C, Fig. 4E and Fig. S2D), we performed differential gene expression analysis among them. As seen in Fig. 4F, natural killer-mediated cytotoxicity was significantly enriched in cluster 4 ($P_{\text{adj}} = 8.4 \times 10^{-9}$) but not significantly enriched in cluster 3 ($P_{\text{adj}} = 0.076$) by KEGG enrichment. Taken together, these results showed that NK-specific gp96 deficiency reduces its maturation and cytotoxic function.”

Also in lines 390-401, “In addition, the significantly increased CD11b⁻CD27⁻ subset in gp96-deficient mice was the most immature NK cells, as this double-negative NK subset transit through less mature CD11b⁻CD27⁺ to CD11b⁺CD27⁺ and CD11b⁺CD27⁻ mature NK cells⁴⁴. Meanwhile, knocking out gp96 in mice showed a similar NK cell development pattern to that of Eomes depletion, and specific and shared transcriptome signatures were observed between gp96-deficient and Eomes-deficient NK cells. In addition, analysis transcript signatures revealed that CD11b/Itgam and CD27 expressions in overall are relatively lower in cluster 3 and higher in cluster 4, in relative to other clusters (Fig. S2B). So there exists apparent overlap between cluster 3 and CD11b⁻CD27⁻ double-negative NK cells, both of which were increased by gp96 KO. So were for cluster 4 and CD11b⁺CD27⁺ DP NK cells, both of which were decreased under gp96 KO.”

To verify whether ILC1s were being inadvertently classified as iNKs, we analyzed percentage of CD127⁺ or CD49a⁺ cells in splenic DX5⁻ NK cells. Please see Fig. S1F-S1G and the results in lines 136-144. “Since NK and ILC1 share many common characteristics in terms of surface markers, we further analyzed ILC1 specific markers CD127 and CD49a expression in splenic DX5⁻ NK cells examine if the defined iNKs contained ILC1s. As can be seen in Fig.S1F and S1G, a small fraction of cells expressed CD127 or CD49a, indicating that a minor amount of ILC1s

were inadvertently classified as iNKs. However, there was no significant difference in the proportion of CD127⁺ and CD49a⁺ cells in CD3⁻CD122⁺NK1.1⁺DX5⁻ cells from *Ncr1^{Cre}*, *gp96^{fl/fl}* and *Ncr1^{Cre}gp96^{fl/fl}* mice, indicating that gp96 KO has no obvious effect on ILC1s.”

As you suggested, we analyzed expression of NKp46 and other key markers in *Ncr1^{Cre}*, *gp96^{fl/fl}* and *Ncr1^{Cre}gp96^{fl/fl}* mice. Please see Fig. S1D-S1E and results in lines 133-136. “To rule out any effects of the *Ncr1*-iCre allele on expressions of NK maturation markers, *Ncr1^{Cre}* mice were used to analyze the key readouts for flow cytometry. *Ncr1^{Cre}* and *gp96^{fl/fl}* mice showed similar expression levels of the marker molecules (Fig. S1D) and percentages of different maturation stages (Fig. S1E).”

And the following sentences have been added to explain the possible reason of abrupt decrease of NKp46 expression. Please see lines 387-390. “Similarly, abruptly decreased NKp46 expression was observed in gp96 KO mice, which may be also due to *Eomes* reduction by gp96 KO as *Eomes* maintains the expression of NKp46 during NK cell development^{15, 43}.”

To address the reason and possible limitation of using DX5 as the upstream gating, the following sentences have been added in the Discussion section. Please see lines 383-387, and lines 415-420. “In addition, *Eomes* is also required for induction of DX5 expression in NK cells¹⁵. We found that gp96 KO resulted in an obviously decreased DX5⁺ mature NK cells by flow cytometry analysis (Fig 1B and 1C), and overexpression of *Eomes* could largely rescue DX5 expression (Fig. 5E). This indicates that gp96 KO-mediated *Eomes* degradation led to DX5 downregulation.”; “As gp96 may regulate DX5 expression via *Eomes*, in current study DX5⁺ was used as an upstream gate flow cytometry. However, much less immature CD11b⁻CD27⁻ NK cells and more DP/CD11b⁺ SP mature NK cells in gp96 KO mice were observed by using CD3⁻NK1.1⁺DX5⁺ gate than using CD3⁻NK1.1⁺ gate, as seen in Fig. 1B and S1C. This phenomenon was not seen in WT mice. Thus, using DX5⁺ as an upstream gate may miss certain types of NK cells that lose DX5 expression.”

Thank you for your comments that greatly strengthened the manuscript.

2. Second, the flow cytometry data in Fig. 5C, which serve as key evidence that gp96-deficient NK cells have reduced Eomes expression, are wholly unconvincing. Indeed, comparison of Eomes expression data in Figs. 5C vs. 6A, suggest that Eomes staining did not work well, if at all, in the experiment depicted in 5C.

Response: As you suggested, we re-analyzed the expression levels of Eomes in NK cells from the spleen and bone marrow using flow cytometry. Isotype was used to determine the efficiency of Eomes staining. Please see revised Fig. 5C. The partly overlap between fluorescence peaks may be due to different Eomes expression in different NK maturation stages.

Thank you.

REVIEWERS' COMMENTS

Reviewer #2 (Remarks to the Author):

The authors addressed most of all my concerns, however, one points as described below should be clarified.

1. In Supplement Fig. 6D, localization of gp96 in whole cell (upper column) and that in cytosol seemed to be almost same, despite that gp 96 is mainly localized within ER. It can be duplicate. However, Fig. 6D showed apparent different signals between whole cell and cytosol could be observed in whole cell and in cytosol even if the authors examined same CD11b+ CD27+ NK cells.

What happened?

2. Another issue I am wondering whetehr the not only ER but also cytosol and nucleus of localization of gp96 is general phenomenon. Please comment on that.

Reviewer #3 (Remarks to the Author):

All major comments/concerns have been addressed.

The authors addressed most of all my concerns, however, one points as described below should be clarified.

1. In Supplement Fig. 6D, localization of gp96 in whole cell (upper column) and that in cytosol seemed to be almost same, despite that gp 96 is mainly localized within ER. It can be duplicate. However, Fig. 6D showed apparent different signals between whole cell and cytosol could be observed in whole cell and in cytosol even if the authors examined same CD11b⁺ CD27⁺ NK cells.

What happened?

Response:

We consider that there may be two reasons why cytosolic and total gp96 signals appear identical in Fig. S6D. Firstly, nucleus gp96 only constitutes a small fraction of the total cellular gp96 protein according to Fig.S6B. Secondly, cytosolic gp96 includes cytoplasm and ER gp96, and ER is the major place that most gp96 resides in. Therefore, subtracting nucleus gp96 doesn't cause significant changes between cytosolic and total gp96 images. Nevertheless, there are subtle differences between total gp96 and cytosolic gp96 images, as indicated by arrows in revised Fig.S6D (Please see revised Fig.S6D). In addition, Fig.S6E was also added to show that only a small fraction of gp96 resides in cell nucleus. Cytosolic gp96 is determined by subtraction of nucleic gp96 (which correlates with nucleic Hoechst staining intensity) from whole cell gp96. The different signals of gp96 of whole cell and cytosol between Supplement Fig. 6D and Fig. 6D may be caused by slightly different expression levels of gp96 among CD11b⁺CD27⁺ NK cells and variations in Hoechst staining intensity across different cells.

The following sentences have been added in lines 400-403 in the Discussion section. "As nucleus gp96 only constitutes a small fraction of the total cellular gp96 protein, and cytosolic gp96 includes cytoplasm and ER gp96, there are only subtle differences between total gp96 and cytosolic gp96 images, as indicated by arrows in Fig.S6D and Fig.S6E."; and in lines 314-316 in the Results section. "Z-stack images also show that only a small fraction of gp96 resides in cell nucleus relative to cytosol that includes cytoplasm as well as ER (Fig.S6E)."; and in lines 489-493 in the Methods section. "Z-stack images of gp96 and Trim28 in whole cell, cytosol and nucleus were generated by Imaris. Before analysis, a new surface for nucleus was created using the Hoechst signal. Nucleic signal and cytosol signal were obtained by the mask channel function in Imaris. Signals retained in the cell nucleus are considered as nuclear gp96 or Trim28, while signals outside the nucleus are regarded as cytosolic gp96 or Trim28."

Thank you for your comments.

2. Another issue I am wondering whetehr the not only ER but also cytosol and nucleus of localization of gp96 is general phenomenon. Please comment on that.

Response:

Although ER is the major place where gp96 resides in, gp96 may also localized in cell nucleus and/or cytosol. Under certain circumstances, gp96 is transported into the nucleus^{1,2}. Meanwhile, gp96 also exists in cell cytosol and interacts with both p53 and

Mdm2 to enhance Mdm2-mediated p53 ubiquitination and degradation³. In addition, cytosol and nucleus localization of gp96 is also inferred by subcellular locations from COMPARTMENTS (<https://www.genecards.org/cgi-bin/carddisp.pl?gene=HSP90B1&keywords=grp94#localization>). COMPARTMENTS localization data are integrated from literature manual curation, high-throughput microscopy-based screens, predictions from primary sequence, and automatic text mining. Confidence scale for cytosol and nucleus localization of gp96 is 5 (dark green), which means high confidence for cytosol and nucleus localization of gp96. The following sentence has been added in lines 394-396 in the Discussion section. “Besides ER where gp96 majorly resides in, this chaperone may also be partly present in the cytoplasm and nucleus to interact with its client proteins^{1, 2, 3}.”

Thank you.

Reference

1. Wu, X.H. *et al.* Dynamic expression of rat heat shock protein gp96 and its gene during development of hepatocellular carcinoma. *Hepatobiliary Pancreat. Dis. Int.* **6**, 616-621 (2007).
2. Yang, Y. & Li, Z. Roles of heat shock protein gp96 in the ER quality control: redundant or unique function? *Mol Cells* **20**, 173-182 (2005).
3. Wu, B. *et al.* Heat shock protein gp96 decreases p53 stability by regulating Mdm2 E3 ligase activity in liver cancer. *Cancer Lett.* **359**, 325-334 (2015).